# Integrability and quench dynamics in the spin-1 central spin XX model

Long Hin Tang[1*], David M. Long[1†], Anatoli Polkovnikov[1], Anushya Chandran[1], Pieter W. Claeys[2]

**1** Department of Physics, Boston University, 590 Commonwealth Ave.,
Boston, MA 02215, USA
**2** Max Planck Institute for the Physics of Complex Systems, 01187 Dresden, Germany
* lhtang@bu.edu, †dmlong@bu.edu

January 16, 2023

## Abstract

Central spin models provide an idealized description of interactions between a central degree of freedom and a mesoscopic environment of surrounding spins. We show that the family of models with a spin-1 at the center and XX interactions of arbitrary strength with surrounding spins is integrable. Specifically, we derive an extensive set of conserved quantities and obtain the exact eigenstates using the Bethe ansatz. As in the homogenous limit, the states divide into two exponentially large classes: *bright* states, in which the spin-1 is entangled with its surroundings, and *dark* states, in which it is not. On resonance, the bright states further break up into two classes depending on their weight on states with central spin polarization zero. These classes are probed in quench dynamics wherein they prevent the central spin from reaching thermal equilibrium. In the single spin-flip sector we explicitly construct the bright states and show that the central spin exhibits oscillatory dynamics as a consequence of the *semilocalization* of these eigenstates. We relate the integrability to the closely related class of integrable Richardson-Gaudin models, and conjecture that the spin-$s_0$ central spin XX model is integrable for any $s_0$.

# 1   Introduction

Central spin models provide a minimal description for a central degree of freedom interacting with an environment of surrounding spins. They are ubiquitous in physics, and have recently gained increased attention with advances in quantum metrology and sensing [1–12]. In such setups the central degree of freedom is typically well controlled and can be used to sense or influence the environment. In solid-state quantum computing platforms, the central degree of freedom could be the spin associated with an electron (hole) in a quantum dot or that associated with a defect center in diamond, while the environment is composed of nuclear spins [13–18]. In cavity-QED systems on the other hand, the cavity acts as the central degree of freedom and the many atoms it interacts with form the environment [19–25].

On the theory side, central spin models have been widely investigated because of their underlying *integrability*. Integrability guarantees an extensive set of conserved quantities and allows all eigenstates to be exactly obtained using Bethe ansatz techniques, which has led to various studies of the equilibrium and dynamical properties in these models [10,26–32]. For XXX interactions ($S_0^x S_j^x + S_0^y S_j^y + S_0^z S_j^z$) between the central spin (at site 0) and the environment spins (at sites $j$), the central spin model belongs to the class of Richardson-Gaudin models [33–36]. Such models are integrable for any value of the central spin and their exact solution has long been established.

In this work, we focus on a system where the central spin interacts with its environment through XX interactions. Such spin-flip terms ($S_0^x S_j^x + S_0^y S_j^y \propto (S_0^+ S_j^- + S_0^- S_j^+)$) naturally arise from dipolar couplings in nuclear magnetic resonance experiments [37–39], in nitrogen vacancy (NV) centers [40], and in certain quantum dots [41]. Some of the authors recently showed that the XX model with a central spin-1/2 particle is integrable with two classes of eigenstates: dark states, in which the central spin is maximally polarized along the $z$-axis and is unentangled with the environment, and bright states, in which the central spin is entangled with the environment [31]. Subsequent work [42] showed that the spin-1/2 XX model remains integrable in the presence of an arbitrarily oriented magnetic field with emergent dark states (building on results in Refs. [31,43,44]). However, in all cases the exact solution strongly depends on the central spin being a spin-1/2 particle.

We here consider the case where a central spin-1 particle interacts with its environment

through XX interactions. By explicitly constructing an extensive set of conserved charges and exact Bethe eigenstates, we establish the integrability of the spin-1 model. While the eigenstate structure is different from that of the spin-1/2 model, we can again identify different classes of bright and dark states. The bright and dark states have striking consequences on the dynamics of the central spin and prevent the central spin from equilibrating with its environment.

This work is structured as follows. In Sec. 2 we present an overview of the integrability of the spin-1 central spin model, detailing its conserved charges and eigenstates. Its integrability can be closely connected to the integrability of XXZ Richardson-Gaudin models, and in Sec. 3 we review some relevant results. These are then used in Sec. 4 to construct the conserved charges and discuss several simple limits. The exact eigenstates are constructed in Sec. 5. These eigenstates are similar to the eigenstates of the homogeneous model (where all couplings are set to be equal), which can be solved in terms of collective spin operators [45]. We therefore present the eigenspectrum for the homogeneous model before moving on to the inhomogeneous model. Following this theoretical analysis, we probe the (semi)localization properties of the eigenstates and the effect on quench dynamics in Sec. 6 in the limiting case of a single spin-flip excitation above the polarized ground state. Dynamics for quenches to resonance from a maximally mixed and unpolarized environment is presented in Sec. 7. We combine the known structure of the eigenstates and the exact solution at the homogeneous point to make predictions for the long-time values of central spin polarization and show that they retain memory of the initial state. Sec. 8 is reserved for conclusions.

While our construction now explicitly depends on the central spin being a spin-1 particle rather than spin-1/2, the integrability of both models suggests that the central spin model with XX interactions is integrable for arbitrary spin values $s_0$. We present two pieces of evidence in support of this conclusion in Appendix C. The first is the integrability of the effective Hamiltonian in the limit of a large $z$-field on the central spin. Specifically, up to second order in the strength of the XX interactions, the effective Hamiltonian obtained by a Schrieffer-Wolff transformation is integrable for any value of the central spin. Second, numerical investigations of the corresponding classical (large $s_0$) model—which can be simulated efficiently—show features of integrability. Integrability of the classical model would imply integrability at smaller $s_0$ within a truncated Wigner approximation [46, 47].

Despite this evidence, proving integrability beyond the spin-1/2 and spin-1 cases remains an outstanding challenge.

## 2   Overview of main results

The focus of this work is the central spin Hamiltonian

$$H = \omega_0' S_0^z + \Omega \sum_{j=1}^{L} S_j^z + \sum_{j=1}^{L} g_j \left( S_0^- S_j^+ + S_0^+ S_j^- \right), \tag{1}$$

describing a central spin-1 particle interacting with an environment of $L$ surrounding spins through an inhomogeneous XX interaction. Both the interaction strengths $g_j$ and the spin quantum numbers $s_j$ of the surrounding environment particles can be chosen freely. The central spin and the environment spins are subject to external fields along the $z$-direction with strength $\omega_0'$ and $\Omega$ respectively. However, since the total $z$ magnetization

$$S_{\text{tot}}^z = \sum_{j=0}^{L} S_j^z, \quad \text{with eigenvalues} \quad M \in \left\{ -1 - \sum_{j=1}^{L} s_j, \ldots, 1 + \sum_{j=1}^{L} s_j \right\}, \tag{2}$$

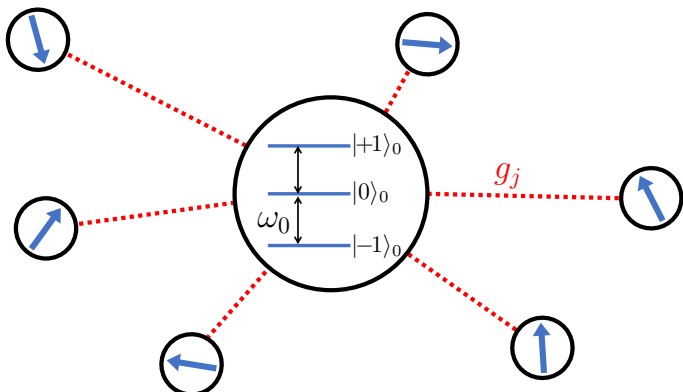

Figure 1: Schematic illustration of the central spin-1 XX Hamiltonian. The central spin-1 particle interacts with an environment of surrounding spins (of any spin quantum number) through an inhomogeneous and anisotropic XX interaction with strength $g_j$. The field strength on the central spin is $\omega_0$.

is conserved, only the detuning $\omega_0 = \omega'_0 - \Omega$ governs the structure of the eigenstates. Indeed, a rotating frame transformation by $e^{-i\Omega S^z_{\text{tot}} t}$ takes $\omega'_0 \mapsto \omega_0$ and $\Omega \mapsto 0$, while leaving the eigenstates (which may be chosen to be eigenstates of $S^z_{\text{tot}}$) unaltered. Without loss of generality, we work within this rotating frame, where the Hamiltonian takes the form

$$H = \omega_0 S^z_0 + \left( S^-_0 G^+ + S^+_0 G^- \right). \tag{3}$$

For convenience, we introduce the environment spin effective raising/lowering operators

$$G^{\pm} = \sum_{j=1}^{L} g_j S^{\pm}_j. \tag{4}$$

This model is illustrated in Fig. 1.

We establish the integrability of the Hamiltonian (1) by constructing both an extensive set of conserved charges and the exact eigenstates. For each environment spin $S_j$ there is an associated conserved charge given by

$$\tilde{Q}_j = \omega_0 S^z_0 Q_j + \omega_0 \left( 2P_0 - 1 \right) S^z_j + \left\{ S^+_0 G^- + S^-_0 G^+, P_0 Q_j \right\}, \tag{5}$$

in which $P_0 = 1 - (S^z_0)^2$ is a projector on central spin $|0\rangle_0$, $\{\cdot, \cdot\}$ the anticommutator, and

$$Q_j = \frac{S^+_j S^-_j + S^-_j S^+_j}{2} + \sum_{k \neq j}^{L} \frac{g_j g_k}{g_j^2 - g_k^2} \left( S^+_j S^-_k + S^-_j S^+_k \right) + 2 \sum_{k \neq j}^{L} \frac{g_k^2}{g_j^2 - g_k^2} S^z_j S^z_k. \tag{6}$$

These conserved charges mutually commute and commute with the central spin Hamiltonian. Note that these charges consist of up to 4-body operators, whereas the conserved charges of the spin-1/2 central spin Hamiltonian consist of up to 2-body operators [31].

Two different classes of exact eigenstates with a fixed number of spin excitations can be constructed by adding excitations to the vacuum state, which is defined as the fully polarized state $|\emptyset\rangle = \otimes_{j=1}^{L} |-s_j\rangle$ if the environment spin at site $j$ has total spin $s_j$. Environment states are then given by unnormalized Bethe states

$$|v_1, \ldots, v_N\rangle = \prod_{a=1}^{N} \left( \sum_{j=1}^{L} \frac{g_j}{g_j^2 - v_a} S^+_j \right) |\emptyset\rangle, \tag{7}$$

parametrized by $N$ (possibly complex) variables $v_1, v_2, \ldots, v_N$. These variables are also known as rapidities[1]. The number of excitations is bounded by $N \leq 2 \sum_{j=1}^{L} s_j$, where equality corresponds to the fully polarized state $\otimes_{j=1}^{L} |s_j\rangle$.

First, the Hamiltonian has a class of degenerate *dark eigenstates*, where the central spin is maximally polarized along either the positive or negative $z$-direction. For $M < 0$, all dark states have central spin down, reading

$$|\mathcal{D}(v_1, \ldots, v_N)\rangle = |-1\rangle_0 \otimes |v_1, \ldots, v_N\rangle \,, \tag{8}$$

with rapidities satisfying a set of Bethe equations

$$\sum_{j=1}^{L} \frac{s_j g_j^2}{g_j^2 - v_a} - \sum_{b \neq a}^{N} \frac{v_b}{v_b - v_a} = 0, \qquad a \in \{1, \ldots, N\} \,, \tag{9}$$

such that $G^- |v_1, \ldots, v_N\rangle = 0$. Note that the rapidities, and thus the dark states $|\mathcal{D}\rangle$, are independent of the magnetic field strength $\omega_0$.

The dark states span a degenerate manifold of energy $E = -\omega_0$:

$$H |\mathcal{D}(v_1, \ldots, v_N)\rangle = -\omega_0 |\mathcal{D}(v_1, \ldots, v_N)\rangle \,. \tag{10}$$

Above half-filling, the dark states have central spin polarization $|+1\rangle_0$. The environment states are annihilated by $G^+$ and can be similarly obtained by spin inversion.

The second class of eigenstates are *bright states*, in which the central spin is entangled with the environment states. Bright states can be parametrized as

$$|\mathcal{B}(\kappa, v_1, \ldots, v_N)\rangle = \sqrt{\frac{1}{2}} |0\rangle_0 \otimes |v_1, \ldots, v_N\rangle$$

$$+ \frac{1}{\kappa - \omega_0} |1\rangle_0 \otimes G^- |v_1, \ldots, v_N\rangle + \frac{1}{\kappa + \omega_0} |-1\rangle_0 \otimes G^+ |v_1, \ldots, v_N\rangle \,,$$

satisfying the eigenvalue equation

$$H |\mathcal{B}(\kappa, v_1, \ldots, v_N)\rangle = \kappa |\mathcal{B}(\kappa, v_1, \ldots, v_N)\rangle \,, \tag{11}$$

provided the rapidities satisfy the set of Bethe equations

$$\frac{\omega_0 - \kappa}{2\kappa} + \sum_{j=1}^{L} \frac{s_j g_j^2}{g_j^2 - v_a} - \sum_{b \neq a}^{N} \frac{v_b}{v_b - v_a} = 0, \qquad \text{for all} \quad a \in \{1, \ldots, N\} \,, \tag{12}$$

$$\kappa(\kappa + \omega_0) = -4 \left( \sum_{a=1}^{N} v_a - \sum_{j=1}^{L} s_j g_j^2 \right) \,. \tag{13}$$

These bright states contain $N + 1$ spin excitations on top of the vacuum state $|-1\rangle_0 \otimes |\emptyset\rangle$ and have total spin magnetization $M = N - \sum_{j=1}^{L} s_j$.

While completeness of the Bethe ansatz is typically not easy to establish, in Sec. 5 we argue that these bright and dark states exhaust all possible eigenstates, such that the Bethe ansatz is complete for this model.

---

[1]Both $v_a$ and $1/v_a$ are used throughout the literature as variables, leading to slightly different Bethe states and equations as compared to e.g. Ref. [31].

# 3 Factorizable Richardson-Gaudin Hamiltonians

In this section, we review various properties of the class of factorizable Richardson-Gaudin Hamiltonians [33–36] that will be useful in establishing the conserved charges and eigenstates of the spin-1 central spin Hamiltonian. The integrability and eigenstates of the spin-1/2 model were similarly obtained using the properties of these models in Ref. [31], but the construction for the spin-1 model is more involved and cannot be seen as a direct generalization of the spin-1/2 model.

The family of factorizable Hamiltonians can be written as

$$H(\alpha) = \frac{1+\alpha}{2} G^+ G^- + \frac{1-\alpha}{2} G^- G^+ = \alpha \sum_{j=1}^{L} g_j^2 S_j^z + \frac{1}{2} \sum_{j,k=1}^{L} g_j g_k \left( S_j^+ S_k^- + S_j^- S_k^+ \right) . \quad (14)$$

The Hamiltonian $H(\alpha)$ is integrable for every choice of $\alpha$, and results for its eigenstates and conserved charges can be found in, for example, Refs. [34, 35, 48–56]. For reference, the conserved quantities are:

$$Q_j(\alpha) = \alpha S_j^z + \frac{S_j^+ S_j^- + S_j^- S_j^+}{2} + \sum_{k \neq j}^{L} \left[ \frac{g_j g_k}{g_j^2 - g_k^2} (S_j^+ S_k^- + S_j^- S_k^+) + \frac{2g_k^2}{g_j^2 - g_k^2} S_j^z S_k^z \right] . \quad (15)$$

These satisfy $[H(\alpha), Q_j(\alpha)] = [Q_j(\alpha), Q_k(\alpha)] = 0$, for all $j, k = 1, \ldots, L$. In the conserved charges of the central spin model we have $Q_j \equiv Q_j(\alpha = 0)$.

All eigenstates can be written as Bethe states, where a Bethe state with $N$ spin excitations on top of the vacuum state $|\emptyset\rangle = \otimes_{j=1}^{L} |-s_j\rangle$ is defined as

$$|v_1, v_2, \ldots v_N\rangle = \prod_{a=1}^{N} G^+(v_a) |0\rangle \qquad \text{with} \qquad G^+(v) = \sum_{j=1}^{L} \frac{g_j}{g_j^2 - v} S_j^+ , \quad (16)$$

expressed in terms of generalized spin raising operators that depend on the parameters $v_1, v_2, \ldots v_N$. These Bethe states are eigenstates provided these rapidities satisfy a set of Bethe equations

$$\frac{\alpha - 1}{2} + \sum_{j=1}^{L} \frac{s_j g_j^2}{g_j^2 - v_a} - \sum_{b \neq a}^{N} \frac{v_b}{v_b - v_a} = 0, \qquad a \in \{1, \ldots, N\}, \quad (17)$$

resulting in eigenvalue equations for the conserved charges,

$$Q_j(\alpha) |v_1, v_2, \ldots v_N\rangle = -2s_j \left[ \frac{\alpha - 1}{2} + \sum_{a=1}^{N} \frac{v_a}{g_j^2 - v_a} - \sum_{k \neq j}^{L} \frac{s_k g_k^2}{g_j^2 - g_k^2} \right] |v_1, v_2, \ldots v_N\rangle , \quad (18)$$

as well as the Hamiltonian,

$$H(\alpha) |v_1, v_2, \ldots v_N\rangle = (\alpha - 1) \left[ \sum_{a=1}^{N} v_a - \sum_{j=1}^{L} s_j g_j^2 \right] |v_1, v_2, \ldots v_N\rangle . \quad (19)$$

Note that the eigenstates and eigenvalues have an implicit dependence on $\alpha$ through the Bethe equations (17). As apparent from Eq. (19), the rapidities can be given an interpretation as spin excitation energies on top of a vacuum energy, with the Bethe equations acting as a set of self-consistency equations.

The model exhibits a quantum phase transition at $|\alpha| = 1$. Consider, for example, $\alpha = 1$, at which point the Hamiltonian reduces to a positive semi-definite Hamiltonian $G^+G^-$. The ground states have energy zero, are necessarily annihilated by $G^-$, and are highly degenerate (see, for example, Ref. [56]). The Bethe ground states are parametrized by $N$ rapidities satisfying

$$\sum_{j=1}^{L} \frac{s_j g_j^2}{g_j^2 - v_a} - \sum_{b \neq a}^{N} \frac{v_b}{v_b - v_a} = 0, \qquad a \in \{1, \ldots, N\} . \tag{20}$$

All excited states have strictly positive energy and correspond to Bethe states with a single diverging rapidity $v \to \infty$, and the remaining $N - 1$ (finite) rapidities satisfy the set of Bethe equations

$$-1 + \sum_{j=1}^{L} \frac{s_j g_j^2}{g_j^2 - v_a} - \sum_{b \neq a}^{N-1} \frac{v_b}{v_b - v_a} = 0, \qquad a \in \{1, \ldots, N-1\} , \tag{21}$$

leading to a strictly positive (energy) eigenvalue $\sum_{j=1}^{L} 2s_j g_j^2 - 2 \sum_{a=1}^{N-1} v_a$ for the Hamiltonian $G^+G^-$. The ground and excited states result in dark and bright states respectively in Sec. 5.

# 4 Conserved charges

The general conserved charges of the central spin Hamiltonian are most easily derived in a $3 \times 3$ block-matrix representation, in which the Hamiltonian (1) is given by

$$H = \begin{pmatrix} \omega_0 & \sqrt{2}G^- & 0 \\ \sqrt{2}G^+ & 0 & \sqrt{2}G^- \\ 0 & \sqrt{2}G^+ & -\omega_0 \end{pmatrix} . \tag{22}$$

The different blocks correspond to different eigenvalues of the central spin polarization $S_0^z$, here ordered as $\{+1, 0, -1\}$, and every matrix element acts on the $L$ environment spins. The diagonal terms correspond to $\omega_0 S_0^z$ and are proportional to the identity within each block. The off-diagonal factors of $\sqrt{2}$ arise from the action of $S_0^{\pm}$ connecting different blocks.

The $L$ corresponding conserved charges (5) establishing integrability can be expressed in block-matrix form as

$$\tilde{Q}_j = \begin{pmatrix} \omega_0(Q_j - S_j^z) & \sqrt{2}G^-Q_j & 0 \\ \sqrt{2}Q_jG^+ & \omega_0 S_j^z & \sqrt{2}Q_jG^- \\ 0 & \sqrt{2}G^+Q_j & -\omega_0(Q_j + S_j^z) \end{pmatrix} . \tag{23}$$

These satisfy $[\tilde{Q}_j, H] = [\tilde{Q}_j, \tilde{Q}_k] = 0$, for all $j, k = 1, \ldots, L$. These properties are checked by direct calculation below using properties of the $Q_j$ as defined in Eq. (6). The different terms in Eq. (23) can first be motivated by considering two simplifying limits.

**Far away from resonance.** Close to the limit $\omega_0 \to \infty$ we can perform a Schrieffer-Wolff transformation [57, 58] to obtain an effective Hamiltonian

$$H_{\text{eff}} = \omega_0 S_0^z + \frac{1}{\omega_0} \left[ S_0^+ G^-, S_0^- G^+ \right] ,$$

$$= \omega_0 S_0^z + \frac{1}{\omega_0} \left( S_0^+ S_0^- G^- G^+ - S_0^- S_0^+ G^+ G^- \right) , \tag{24}$$

which has block-matrix representation

$$
\begin{aligned}
H_{\text{eff}} &= \begin{pmatrix} \omega_0 + \frac{2}{\omega_0} G^- G^+ & 0 & 0 \\ 0 & \frac{2}{\omega_0}[G^-, G^+] & 0 \\ 0 & 0 & -\omega_0 - \frac{2}{\omega_0} G^+ G^- \end{pmatrix} \\
&= \begin{pmatrix} \omega_0 + \frac{2}{\omega_0} H(\alpha = -1) & 0 & 0 \\ 0 & -\frac{4}{\omega_0} H(\alpha \to \infty) & 0 \\ 0 & 0 & -\omega_0 - \frac{2}{\omega_0} H(\alpha = 1) \end{pmatrix}
\end{aligned}
\tag{25}
$$

All diagonal elements correspond to Richardson-Gaudin integrable Hamiltonians for the environment from Sec. 3, such that the effective Hamiltonian itself is also integrable. The diagonal elements of Eq. (23) are the dominant terms for $\omega_0 \to \infty$ and correspond exactly to the conserved charges of the Hamiltonian in Eq. (25).

**At resonance.** In the opposite limit where $\omega_0 = 0$, i.e. at resonance, the diagonal elements of both the Hamiltonian (22) and the conserved charges (23) vanish. In this limit the commutator of the Hamiltonian with the conserved charges can be directly evaluated as

$$
[H, \tilde{Q}_j] = \begin{pmatrix} 0 & 0 & 0 \\ 0 & 2[G^+ G^- + G^- G^+, Q_j] & 0 \\ 0 & 0 & 0 \end{pmatrix}.
\tag{26}
$$

This expression for the commutator is independent of the choice of $Q_j$. The single nontrivial element now vanishes since $G^+ G^- + G^- G^+$ is again a Richardson-Gaudin integrable Hamiltonian, with conserved charges $Q_j$.

An alternative way of obtaining the conserved charge in this limit is by noting that $H^2$ commutes with $(S_0^z)^2$. If we consider the component of $H^2$ in the space with zero spin polarization, we have that

$$
P_0 H^2 = 2 P_0 \left( G^+ G^- + G^- G^+ \right),
\tag{27}
$$

returning the factorizable Hamiltonian with conserved charges $Q_j$ for the environment space. The Hamiltonian squared then has conserved charges $P_0 Q_j$, such that at resonance the Hamiltonian itself has conserved charges $\{H, P_0 Q_j\}$. Expressing these charges as a block matrix then returns the conserved charges (23) with $\omega_0 = 0$.

**General.** The general conserved charges (23) are linear in $\omega_0$, such that they interpolate between the two limiting cases. The commutation relation at arbitrary values of $\omega_0$ can be checked by evaluating the commutator $[H, \tilde{Q}_j]$, which reads

$$
\begin{pmatrix} 0 & \sqrt{2}\omega_0(G^- Q_j^{(+)} - Q_j^{(-)} G^-) & 0 \\ \sqrt{2}\omega_0(G^+ Q_j^{(-)} - Q_j^{(+)} G^+) & 2[G^+ G^- + G^- G^+, Q_j] & -\sqrt{2}\omega_0(G^- Q_j^{(+)} - Q_j^{(-)} G^-) \\ 0 & -\sqrt{2}\omega_0(G^+ Q_j^{(-)} - Q_j^{(+)} G^+) & 0 \end{pmatrix}
\tag{28}
$$

where we have introduced the shorthand $Q_j^{(\pm)} = Q_j \pm S_j^z$. No properties of the $Q_j$ have been used yet. The diagonal element again vanishes since $Q_j$ are the conserved charges of $G^+ G^- + G^- G^+$ [see Eq. (15)]. The off-diagonal elements can be shown to vanish by noting that $G^+ Q_j^{(-)} = Q_j^{(+)} G^+$ or $G^- Q_j^{(+)} = Q_j^{(-)} G^-$ (see, for example, Ref. [31]).

It is an open question how the integrability of this model and the construction of the conserved charges can be incorporated in the general framework of (Richardson-Gaudin) integrability such as generalized Gaudin algebras [49], the algebraic Bethe ansatz [59], or constructions based on solutions to the "generalized" classical Yang-Baxter equation [44,60].

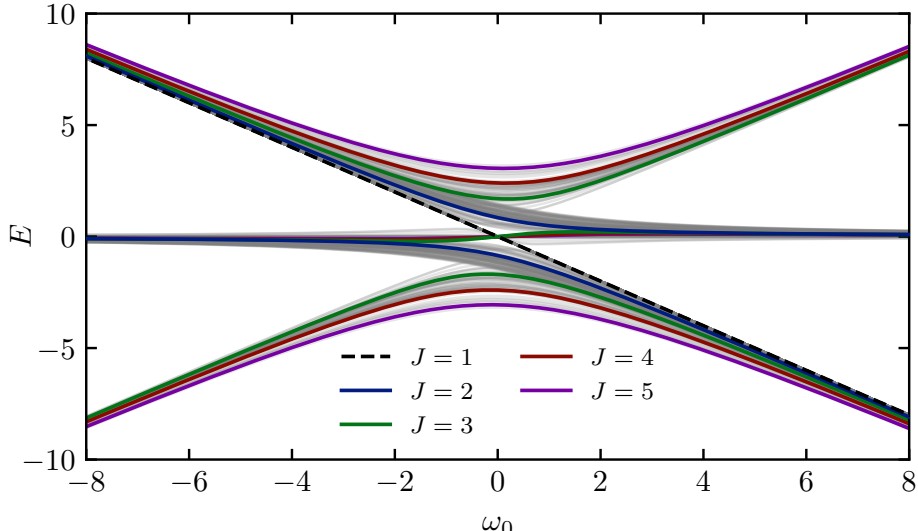

Figure 2: Illustration of the spectrum of the central spin Hamiltonian (with all environment spins being spin-1/2) in the homogeneous case (dashed black line and full colored lines) and the inhomogeneous case (full gray lines) in the sector with $L = 10$, $N = 4$ such that $M = -2$. In the homogeneous case, dark states correspond to states with environment spin $J = |M| - 1 = 1$. Sectors with $J = |M| = 2$ contribute two states, sectors with $|M| < J \leq L/2 = 5$ contribute three states.

## 5 Eigenstates

Central spin XX models generally support two different classes of eigenstates: bright and dark states. All such states can be obtained explicitly and systematically. We note that the construction of dark states does not depend on the value of the central spin (see, for example, Refs. [45, 61–64]), such that the construction for dark states in the spin-1/2 model immediately extends to the current model. The construction of the bright states, however, is particular to this model, and these exhibit a richer behavior as compared to the spin-1/2 case.

### 5.1 Homogeneous limit

It is instructive to first examine the special case of homogeneous couplings, $g_j = g$ for all $j$. In this case $G^{\pm}$ is proportional to the total spin raising/lowering operator on the environment and we can write

$$H = \omega_0 S_0^z + g \left( S_0^+ J^- + S_0^- J^+ \right), \quad \text{where} \quad J^\mu = \sum_{j=1}^{L} S_j^\mu. \tag{29}$$

The eigenstates of the model (29) can be found without resorting to Bethe ansatz machinery. The homogeneous limit has symmetries

$$[H, S_0^z + J^z] = [H, S_0^2] = [H, J^2] = 0, \tag{30}$$

where $J^2 = \frac{1}{2}(J^+ J^- + J^- J^+) + (J^z)^2$ is the total spin operator for the environment. The central spin Hamiltonian can be represented as a block-diagonal matrix in a fixed $(M, J)$

sector of linear dimension 1, 2 or 3, depending on the relation between $J$ and $M$. All such matrices can be explicitly diagonalized to return the spectrum of the homogeneous central spin model.

**Bright states**. We first consider the case $J > |M|$. The Hamiltonian has contributions from $3 \times 3$ blocks spanned by

$$|+1\rangle_0 \otimes |J, M - 1\rangle, \qquad |0\rangle_0 \otimes |J, M\rangle, \qquad |-1\rangle_0 \otimes |J, M + 1\rangle, \tag{31}$$

where $|m_0\rangle_0$ denotes the eigenstates of $S_0^z$ and $|J, M_J\rangle$ is a simultaneous eigenstate of $J^2$ and $J^z$ with eigenvalues $J(J + 1)$ and $M_J$, respectively. In this block $H$ takes the form

$$H_{JM} = \begin{pmatrix} \omega_0 & \sqrt{2} g c_{JM}^- & 0 \\ \sqrt{2} g c_{JM}^- & 0 & \sqrt{2} g c_{JM}^+ \\ 0 & \sqrt{2} g c_{JM}^+ & -\omega_0 \end{pmatrix} \quad \text{with} \quad c_{JM}^\pm = \sqrt{(J \mp M)(J \pm M + 1)}. \tag{32}$$

The resulting states generally depend strongly on $\omega_0$ and are known as *bright states*—to be contrasted with the dark states that will be introduced later in this section.

In the special case of resonance ($\omega_0 = 0$) the eigenstates can be analytically constructed and used to further divide the bright states in two subclasses. The Hamiltonian matrix reduces to

$$H_{JM} = \sqrt{2} g \begin{pmatrix} 0 & c_{JM}^- & 0 \\ c_{JM}^- & 0 & c_{JM}^+ \\ 0 & c_{JM}^+ & 0 \end{pmatrix}. \tag{33}$$

A single eigenstate can be constructed as

$$|\mathcal{B}_0\rangle = \frac{1}{B} \left( c_{JM}^+ |+1\rangle_0 \otimes |J, M - 1\rangle - c_{jM}^- |-1\rangle_0 \otimes |J, M + 1\rangle \right), \tag{34}$$

with $|B|^2 = (c_{JM}^+)^2 + (c_{JM}^-)^2$ and energy $E_0 = 0$. A notable feature of this class is that these states have no weight on $|0\rangle_0$. This feature will persist to the inhomogeneous case, and we will refer to these states as *double states*.

The remaining two eigenstates follow as

$$|\mathcal{B}_\pm\rangle = \frac{1}{\sqrt{2}} |0\rangle_0 \otimes |J, M\rangle \pm \frac{1}{\sqrt{2} B} \left( c_{JM}^- |+1\rangle_0 \otimes |J, M - 1\rangle + c_{JM}^+ |-1\rangle_0 \otimes |J, M + 1\rangle \right), \tag{35}$$

with degenerate energy eigenvalues $E_\pm = \sqrt{2} g B$. The persistent feature of this class is that exactly half the weight of the state is on $|0\rangle_0$. These states form pairs and since they always have support on all three central spin states we will refer to these as *triple states*.

Next, we can consider the case $J = |M|$ with $M \neq 0$. The Hamiltonian now only couples two different states, depending on the sign of $M$,

$$|0\rangle_0 \otimes |M, M\rangle \quad \text{and} \quad |+1\rangle_0 \otimes |M, M - 1\rangle \qquad \text{for} \qquad M > 0, \tag{36}$$
$$|0\rangle_0 \otimes |-M, M\rangle \quad \text{and} \quad |-1\rangle_0 \otimes |-M, M + 1\rangle \qquad \text{for} \qquad M < 0. \tag{37}$$

Constructing the central spin Hamiltonian in this basis leads to a $2 \times 2$ matrix. For example, for $M > 0$,

$$H_{MM} = \begin{pmatrix} 0 & 2g\sqrt{M} \\ 2g\sqrt{M} & \omega_0 \end{pmatrix}, \tag{38}$$

which can be explicitly diagonalized to return a pair of eigenvalues

$$E = \frac{\omega_0}{2} \pm \sqrt{\frac{\omega_0^2}{4} + 4Mg^2} \,. \tag{39}$$

These states have common properties of both double and triple states. At resonance the corresponding eigenstates are supported on two states, as with double states, but half the weight of the state is on $|0\rangle_0$, as with the triple states. Since we will be interested in quench dynamics of the central spin magnetization, we also refer to these as *triple states*.

**Dark states.** If we consider a total environment spin where $J = |M| - 1$, the blocks reduce to $1 \times 1$ blocks. This condition enforces that the environment is in a state $|J, J\rangle$ or $|J, -J\rangle$, and the corresponding states can now only take the form

$$|+1\rangle_0 \otimes |J, J\rangle = |+1\rangle_0 \otimes |M - 1, M - 1\rangle \qquad \text{for} \qquad M > 0, \tag{40}$$
$$|-1\rangle_0 \otimes |J, -J\rangle = |-1\rangle_0 \otimes |-M - 1, M + 1\rangle \qquad \text{for} \qquad M < 0. \tag{41}$$

Crucially, these states have the property that they are annihilated by both $S_0^+ J^-$ and $S_0^- J^+$, the interaction terms in the Hamiltonian, such that $|\mathcal{D}\rangle = |\pm 1\rangle_0 \otimes |J, \pm J\rangle$ is an eigenstate of $H$ with eigenvalue $\pm \omega_0$. These product eigenstates $|\mathcal{D}\rangle$ are called *dark states* and are independent of $\omega_0$. Note that dark states with central spin state $|+1\rangle_0$ only appear for $M > 0$, whereas dark states with central spin state $|-1\rangle_0$ only appear for $M < 0$. In the specific case where $J = M = 0$, the homogeneous model supports additional dark states of the form $|0\rangle_0 \otimes |0, 0\rangle$, which are eigenstates of the Hamiltonian with zero eigenvalue.

In Appendix A, we provide the counting of each class of states, assuming each environmental spin is spin-1/2. In the limit of large $L$, there are twice as many triple states as double states, while the ratio of the number of dark states to that of triple states scales as $|M|/L$. We use these results to predict the late-time expectation value of central spin projectors in Sec. 7.

## 5.2 The inhomogeneous model

The eigenstates of the inhomogeneous model can be constructed as Bethe states that are a direct generalization of the eigenstates of the homogeneous model.

**Dark states**

At every value of the magnetic field $\omega_0$, the central spin Hamiltonian in Eq. (1) supports a set of dark eigenstates in which the central spin is not entangled with the environment spins. These dark states $|\mathcal{D}\rangle$ are product states of the form $|-1\rangle_0 \otimes |\mathcal{D}^-\rangle$ or $|+1\rangle_0 \otimes |\mathcal{D}^+\rangle$, where the environment state is adiabatically connected to the states $|J, \pm J\rangle$ and satisfies [31,45,61–63]

$$G^\pm |\mathcal{D}^\pm\rangle = 0. \tag{42}$$

This condition guarantees that such dark states are annihilated by the (inhomogeneous) interaction part of the Hamiltonian (1), as well as being eigenstates of the central spin term $S_0^z$ with eigenvalues $\pm 1$. As such, dark states are independent of the central spin field $\omega_0$ and form degenerate manifolds with energy $\pm \omega_0$.

As outlined in Ref. [31], the environment states correspond to the ground states of the factorizable Hamiltonians $G^+ G^-$ and $G^- G^+$, which can be expressed as Bethe states satisfying the Bethe equations (20). As reviewed in Sec. 3, for $M < 0$ these dark states can be written as

$$|\mathcal{D}(v_1, \ldots, v_N)\rangle = |-1\rangle_0 \otimes |v_1, \ldots, v_N\rangle \,, \tag{43}$$

with rapidities satisfying the Bethe equations

$$\sum_{j=1}^{L} \frac{s_j g_j^2}{g_j^2 - v_a} - \sum_{b \neq a}^{N} \frac{v_b}{v_b - v_a} = 0, \qquad a \in \{1, \dots, N\}. \tag{44}$$

**Bright states**

The bright eigenstates can similarly be related to the eigenstates of the factorizable Richardson-Gaudin models, albeit in a more involved way. Specifically, we consider an ansatz expressed in terms of a single environment state $|\psi\rangle$ and a free parameter $\kappa$, writing

$$|\mathcal{B}\rangle = \sqrt{\frac{1}{2}} |0\rangle_0 \otimes |\psi\rangle + \frac{1}{\kappa - \omega_0} |1\rangle_0 \otimes G^- |\psi\rangle + \frac{1}{\kappa + \omega_0} |-1\rangle_0 \otimes G^+ |\psi\rangle. \tag{45}$$

The above state is an eigenstate of the Hamiltonian (1) with eigenvalue $\kappa$ provided the environment state $|\psi\rangle$ satisfies the (self-consistent) eigenvalue equation

$$H_\kappa |\psi\rangle = \left[ \frac{G^+ G^-}{\kappa - \omega_0} + \frac{G^- G^+}{\kappa + \omega_0} \right] |\psi\rangle = \frac{\kappa}{2} |\psi\rangle. \tag{46}$$

This equation is self-consistent because the Hamiltonian $H_\kappa$ depends on the eigenvalue, but, crucially, is Richardson-Gaudin integrable for every choice of $\kappa$. As such, its eigenstates can be exactly constructed as Bethe states for every choice of $\kappa$. The Hamiltonian is (up to a prefactor) the Hamiltonian from Eq. (14) from Sec. 3, where the parameter $\alpha$ can be determined as $\omega_0/\kappa$. In order to find a set of Bethe equations we can express the eigenvalue $\kappa$ in terms of rapidities, and now the Bethe equations for these rapidities need to be modified to take into account the self-consistency. This approach directly returns the equations (12).

**Spectrum at resonance**

The eigenstates of the inhomogeneous model at resonance can again be compared with the eigenstates at resonance in the homogeneous limit, recovering the double, triple and dark states.

At resonance, $\omega_0 = 0$, the self-consistency equation for the bright states (46) reduces to a regular eigenvalue equation. In this limit the self-consistent equation can be rewritten as

$$\left[ G^+ G^- + G^- G^+ \right] |\psi\rangle = \frac{\kappa^2}{2} |\psi\rangle, \tag{47}$$

such that the environment states will correspond to the eigenstates of the above (integrable) Hamiltonian. Since the Hamiltonian $G^+ G^- + G^- G^+$ is positive definite, $\kappa^2$ is always positive. For a given eigenstate of this Hamiltonian with eigenvalue $\kappa^2/2$, the central spin Hamiltonian has two corresponding eigenstates with eigenvalue $\pm\kappa$, given by

$$|\mathcal{B}_\pm\rangle = \sqrt{\frac{1}{2}} |0\rangle_0 \otimes |\psi\rangle \pm \frac{1}{\kappa} \left( |+\rangle_0 \otimes G^- |\psi\rangle + |-\rangle_0 \otimes G^+ |\psi\rangle \right). \tag{48}$$

These are the (normalized) *triple states* identified previously in the homogeneous limit, and continue to have exactly half their weight on $|0\rangle_0$.

The *double states*, with zero energy and vanishing weight on $|0\rangle_0$, can be constructed in an alternative way (since in this limit the corresponding Bethe equations become singular):

$$|\mathcal{B}_0\rangle = |-\rangle_0 \otimes G^+ (G^- G^+)^{-1} |\psi\rangle - |+\rangle_0 \otimes G^- (G^+ G^-)^{-1} |\psi\rangle, \tag{49}$$

where the inverse should be interpreted as a pseudo-inverse, with the condition that the pseudo-inverse should act as the actual inverse on the environment states $|\psi\rangle$, i.e.

$$G^- G^+ (G^- G^+)^{-1} |\psi\rangle = G^+ G^- (G^+ G^-)^{-1} |\psi\rangle = |\psi\rangle \,. \tag{50}$$

This condition can be satisfied if we consider an initial state that has vanishing overlap with the dark states, since the dark states lie in the kernel of either $G^+ G^-$ or $G^- G^+$. For example, if we consider $M < 0$, all dark states are annihilated by $G^+ G^-$ whereas $G^- G^+$ has no dark states, such that the inverse of $G^- G^+$ is well defined. Taking the state $|\psi\rangle$ to be an excited state of $G^+ G^-$, every excited state gives rise to a well defined double state.

**Completeness**

It is possible to count the total number of dark and bright states and show that they exhaust all eigenstates of the central spin Hamiltonian (1). The number of these states depends on the choice of environment spins, and for concreteness we here focus on the case where each environmental spin is spin-1/2 and $M < 0$. The argument for completeness does not depend on the specific choice of spins or total magnetization.

The total number of dark states is set by the number of solutions to

$$G^- |\mathcal{D}^-\rangle = 0 \,. \tag{51}$$

For a total magnetization $M$ and a dark state $|-1\rangle_0 \otimes |\mathcal{D}^-\rangle$, the state $|\mathcal{D}^-\rangle$ has a magnetization $M + 1$ and the state $G^- |\mathcal{D}^-\rangle$ has magnetization $M$. The total number of solutions to the above equation is given by the dimension of the former Hilbert space (fixing the number of variables) minus the dimension of the latter (fixing the number of constraints), and we find[2]

$$N_{\text{dark}} = \binom{L}{M + L/2 + 1} - \binom{L}{M + L/2}. \tag{52}$$

The same result can be recovered in the homogeneous case (see Eq. (84)).

For the bright states, the total number of solutions to the self-consistent equation can be found by plotting the spectrum of the Hamiltonian

$$H_\kappa = \frac{G^+ G^-}{\kappa - \omega_0} + \frac{G^- G^+}{\kappa + \omega_0} \tag{53}$$

as a function of $\kappa$, as illustrated in Fig. 3 for generic choices of the interaction strengths and $\omega_0$. Our conclusions do not depend on any specific choice of the parameters.

Any intersection between this spectrum and the dashed red line denoting $\kappa$ determines a solution to the self-consistent equation (46). The number of solutions can now be directly related to the number of bright states: the different lines in Fig. 3 correspond to the different eigenstates of the Hamiltonian (53). As $\kappa \to \pm\infty$ all eigenvalues go to zero. At intermediate values of $\kappa$ all eigenvalues are monotonously decreasing, as follows from the Hellmann-Feynman theorem:

$$\frac{\partial E}{\partial \kappa} = \left\langle \frac{\partial H_\kappa}{\partial \kappa} \right\rangle = -\frac{\langle G^+ G^- \rangle}{(\kappa - \omega_0)^2} - \frac{\langle G^- G^+ \rangle}{(\kappa + \omega_0)^2} \leq 0. \tag{54}$$

Here we have made use of the positive semi-definiteness of $G^\pm G^\mp$. Since the Hamiltonian

---

[2]Eq. (52) requires that $G^-$ is surjective on the $M$ magnetization sector. This can be seen by noting that $G^- = P^{-1} J^- P$ is related to the total spin lowering operator by a similarity transformation, where $P = \exp\left(\sum_{j=1}^{L} -\ln g_j S_j^z\right)$ [56].

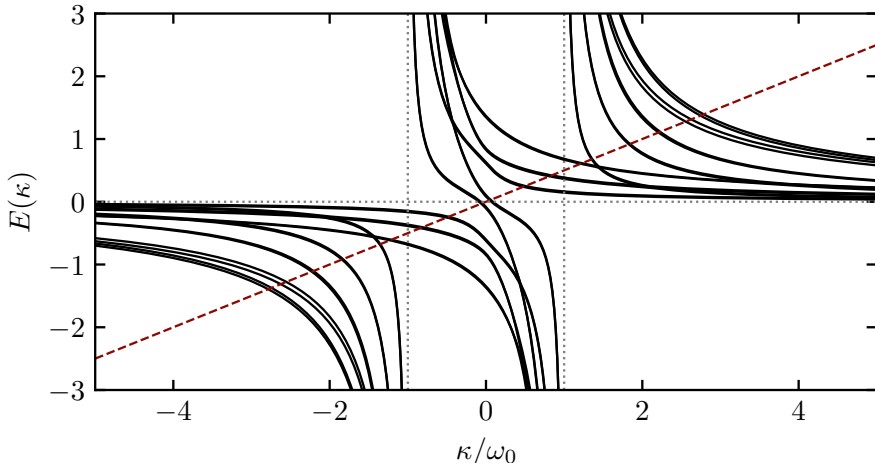

Figure 3: Graphical illustration of the self-consistency equation for $L = 4$. Full black lines denote the spectrum $E(\kappa)$ of the Hamiltonian (53) and the dashed red line $E(\kappa) = \kappa/2$ (46). Vertical dotted lines mark the asymptotics at $\kappa = \pm\omega_0$.

diverges at $\kappa = \pm\omega_0$, there are now two options for every eigenvalue $E(\kappa)$: either these diverge at $\kappa = \pm\omega_0$ and the eigenvalue has two vertical asymptotes, or the corresponding eigenstate is annihilated by the residue of $H_\kappa$ at $\kappa = \omega_0$ or $\kappa = -\omega_0$ and the eigenvalue has a single vertical asymptote (the state cannot be annihilated by both, as will be made apparent shortly). In the former case the eigenvalue has 3 intercepts with the diagonal line, leading to 3 bright state solutions per environment state. In the latter case the eigenvalue has 2 intercepts with the diagonal line, leading to 2 bright states per environment state. Crucially, the number of states that are annihilated by the residue is exactly equal to the number of dark states, since these are the states that are annihilated by $G^+G^-$ (or $G^-G^+$ for $M > 0$).

As such, the total number of bright state solutions equals three times the environment space dimension minus the number of dark states, which combined with the total number of dark states returns the full dimension of the spin-1 central spin Hamiltonian, where the central spin can take 3 different values. The completeness of the Bethe ansatz for the spin-1 central spin Hamiltonian then directly follows from the completeness of the Bethe ansatz for the Richardson-Gaudin Hamiltonian (14) [65, 66].

# 6   Single excitation

In the case of a single spin excitation the eigenstates are amenable to a more detailed analytical treatment, even away from resonance and in the limit of an infinite environment $L \to \infty$. In the following, we first analyze the localization properties of the (bright) eigenstates and then present exact results for quench dynamics starting from a product state.

## 6.1   Multifractality and semilocalization

In the case of a single excitation, the Hamiltonian (1) can be written as a so-called arrowhead matrix. Such models generally support both dark and bright eigenstates, and these states have recently gained attention [67] in the context of *semilocalization*, being neither fully localized nor fully delocalized. Calculations of the inverse participation ratio

(IPR) instead indicated a multifractal behavior.

The IPR is defined as

$$\mathcal{P}(q) = \sum_{j=0}^{L} |\psi_j|^{2q}, \tag{55}$$

with $|\psi_j|^2$ the component of the (normalized) wave function where the excitation is located on spin $j$. For a delocalized eigenstate, all components are on the order $1/L$, resulting in an IPR scaling with $L$ as $\mathcal{P}(q) = \mathcal{O}(L^{1-q})$, whereas a localized eigenstate has a few components $\mathcal{O}(1)$, resulting in a scaling of $\mathcal{P}(q) = \mathcal{O}(1)$. A change of scaling as $q$ is varied is a signature of multifractality in the eigenstate [67–70].

For a single excitation in the central spin model, there are $L - 1$ dark states and 2 bright states. In the construction of the bright states (45), the environment state $|\psi\rangle$ is necessarily the vacuum state $|\emptyset\rangle$. The wave function reads

$$|\kappa\rangle = \sqrt{\frac{1}{2}} |0\rangle_0 \otimes |\emptyset\rangle + \frac{1}{\kappa + \omega_0} |-1\rangle_0 \otimes G^+ |\emptyset\rangle, \tag{56}$$

with $\kappa$ satisfying

$$\left[ \frac{G^+ G^-}{\kappa - \omega_0} + \frac{G^- G^+}{\kappa + \omega_0} \right] |\emptyset\rangle = \frac{G^- G^+}{\kappa + \omega_0} |\emptyset\rangle = \frac{\kappa}{2} |\emptyset\rangle. \tag{57}$$

As $G^- |\emptyset\rangle = 0$ and $G^- G^+ |\emptyset\rangle = [G^-, G^+] |\emptyset\rangle = 2 \sum_{j=1}^{L} s_j g_j^2 |\emptyset\rangle$, the self-consistent eigenvalue equation simplifies to a quadratic equation for $\kappa$. This quadratic equation can be explicitly solved to return the two bright states.

In order to have a finite $\kappa$ value when the number of environment sites $L$ goes to infinity, we will consider a distribution of interaction strengths $g_j = \tilde{g}_j/\sqrt{L}$ with $\tilde{g}_j$ distributed in some fixed interval. The quadratic equation returns two solutions corresponding to two bright states with

$$\kappa = -\frac{\omega_0}{2} \pm \sqrt{\frac{\omega_0^2}{4} + 2\overline{g}^2} \qquad \text{with} \qquad \overline{g}^2 = \frac{1}{L} \sum_{j=1}^{L} 2 s_j \tilde{g}_j^2. \tag{58}$$

Crucially, $\kappa$ stays finite in the limit $L \to \infty$, resulting in both a finite eigenvalue and nonzero components in the wave function (56). The normalized components of the wave function immediately follow as

$$|\psi_0|^2 = \frac{(\kappa + \omega_0)^2}{(\kappa + \omega_0)^2 + 2\overline{g}^2}, \qquad |\psi_j|^2 = 2\frac{2 s_j g_j^2}{(\kappa + \omega_0)^2 + 2\overline{g}^2}, \quad j = 1, \ldots, L. \tag{59}$$

The IPR can be calculated from these components as

$$\mathcal{P}(q) = |\psi_0|^{2q} + \sum_{j=1}^{L} |\psi_j|^{2q} = \frac{(\kappa + \omega_0)^{2q}}{((\kappa + \omega_0)^2 + 2\overline{g}^2)^q} + \frac{2^q}{L^q} \sum_{j=1}^{L} \left( \frac{2 s_j \tilde{g}_j^2}{(\kappa + \omega_0)^2 + 2\overline{g}^2} \right)^q. \tag{60}$$

Assuming a uniform distribution for $2 s_j \tilde{g}_j^2$ in a finite interval $[0, 2\overline{g}^2]$, the sum can be explicitly evaluated to return

$$\mathcal{P}(q) = \frac{(\kappa + \omega_0)^{2q}}{((\kappa + \omega_0)^2 + 2\overline{g}^2)^q} + \frac{1}{L^{q-1}} \frac{4^q \overline{g}^{2q}}{(q+1)((\kappa + \omega_0)^2 + 2\overline{g}^2)^q}. \tag{61}$$

The scaling of the IPR with $L$ for a fixed $q$ results in

$$\mathcal{P}(q) = \begin{cases} \mathcal{O}(L^{1-q}) & \text{if} \quad 0 < q < 1, \\ \mathcal{O}(1) & \text{if} \quad 1 < q. \end{cases} \tag{62}$$

This quantifies what is already apparent from the parametrizations (56) and (59): in the thermodynamic limit the component of the bright states on the central spin remains $\mathcal{O}(1)$, whereas all other components are delocalized over the environment states and $\mathcal{O}(1/L)$. This scenario has been dubbed *semilocalization* [67, 70].

## 6.2   Quench dynamics

The effect of semilocalization can be directly observed in quench dynamics. We consider quenches where the system is initially prepared in a product state, with the single excitation localized either on the central spin or on one of the environment spins, and is subsequently evolved using the central spin Hamiltonian.

For simplicity, we focus on the dynamics of the central spin magnetization $\langle S_0^z(t) \rangle$ starting from a general initial state $|\psi_0\rangle$. Since the dark states are eigenstates of $S_0^z$ all nontrivial dynamics is due to the two bright states, and we can write

$$\langle S_0^z(t) \rangle = -\sum_{\mathcal{D}} |\langle \psi_0 | \mathcal{D} \rangle|^2 + \sum_{\kappa = \kappa_\pm} \langle \kappa | S_0^z | \kappa \rangle |\langle \kappa | \psi_0 \rangle|^2$$
$$+ \left( e^{-i(\kappa_+ - \kappa_-)t} \langle \psi_0 | \kappa_- \rangle \langle \kappa_- | S_0^z | \kappa_+ \rangle \langle \kappa_+ | \psi_0 \rangle + \text{h.c.} \right), \tag{63}$$

where we have labeled the two bright states by their eigenvalues $\kappa_\pm = -\omega_0/2 \pm \sqrt{\omega_0^2/4 + 2\overline{g}^2}$ and the $(L-1)$ dark states as $\mathcal{D}$. The central spin magnetization oscillates with a single frequency $\kappa_+ - \kappa_- = 2\sqrt{\omega_0^2/4 + 2\overline{g}^2}$, and both the amplitude of the oscillations and their average value are determined by the overlaps with the bright states.

Consider first the case where the initial state consists of an excitation on the central spin, i.e. $|\psi_0\rangle = |0\rangle_0 \otimes |\emptyset\rangle$. This state has a vanishing overlap with the dark states, and the contribution from the bright states can be calculated using the explicit parametrization (56) as

$$\langle \kappa | S_0^z | \kappa \rangle |\langle \kappa | \psi_0 \rangle|^2 = -\frac{\overline{g}^2}{2\left(\omega_0^2/4 + 2\overline{g}^2\right)}, \tag{64}$$

which holds for both bright states $|\kappa_\pm\rangle$. The resulting central spin dynamics immediately follows as

$$\langle S_0^z(t) \rangle = -\frac{\overline{g}^2}{\omega_0^2/4 + 2\overline{g}^2} \left[ 1 - \cos\left( 2t\sqrt{\omega_0^2/4 + 2\overline{g}^2} \right) \right], \tag{65}$$

This result is illustrated in Fig. 4 and is identical to the dynamics in the homogeneous model (29) with interaction strength $g = \overline{g}/\sqrt{2J}$ and environment spin $J = \sum_{j=1}^{L} s_j$. Crucially, the amplitude of the central spin oscillation remains finite in the limit $L \to \infty$ provided $\overline{g}^2$ remains finite. The nonvanishing amplitude of the oscillation is a direct consequence of the semilocalized nature of the bright states: the overlap between the initial state and the two bright states remains $\mathcal{O}(1)$ in this limit.

These dynamics can be contrasted with the central spin dynamics for an initial product state localized on an environment spin. The amplitude of the central spin oscillations is set by $\langle \psi_0 | \kappa_- \rangle \langle \kappa_+ | \psi_0 \rangle$ and hence by the component $\psi_j$ from Eq. (59) for an initial excitation localized on spin $j$. The individual overlaps scale as $\mathcal{O}\left(1/\sqrt{L}\right)$, such that the

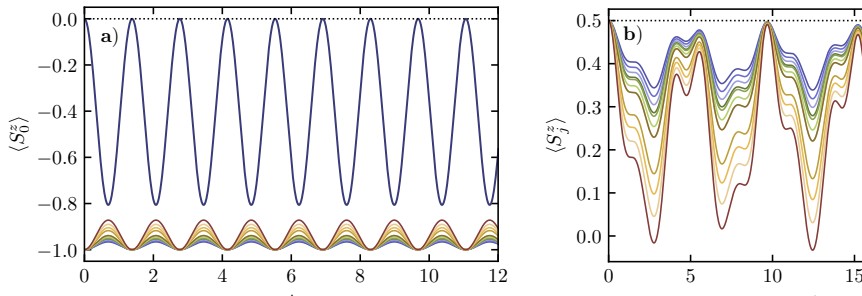

Figure 4: Quench dynamics in single-excitation sector away from resonance. (**a**) Dynamics of the central spin magnetization $S_0^z$ for an initial product state that is either localized on the central spin (line starting at $\langle S_0^z \rangle = 0$) or on an environment spin, with the different lines starting at $\langle S_0^z \rangle = -1$ illustrating the $L$ different initial states. (**b**) Dynamics of the environment spin magnetization $S_j^z$ for an excitation initially localized on site $j$. *Parameters:* $L = 12$, $\omega_0 = 2$, and $\tilde{g}_j$ is uniformly distributed in the interval $[1, 2]$ for environment spins with $s_j = 1/2$.

total amplitude of the spin oscillations will scale as $\mathcal{O}(1/L)$. As illustrated in Fig. 4(a) and Fig. 5(a), central spin oscillations are indeed suppressed for states initially localized in the environment.

A similar behavior is observed in the dynamics of the environment spin polarizations $\langle S_j^z(t) \rangle$ for an excitation initially localized on spin $i$. Since the dark states are no longer eigenstates of the observable these will also contribute to the dynamics, and the spins will oscillate with three frequencies

$$\kappa_+ - \kappa_- = 2\sqrt{\omega_0^2/4 + 2\overline{g}^2}, \qquad \kappa_\pm + \omega_0 = \omega_0/2 \pm \sqrt{\omega_0^2/4 + 2\overline{g}^2}\,. \qquad (66)$$

The dynamics is illustrated in Fig. 4(b). Note that the frequencies are generally not commensurate, but the spin dynamics may exhibit *approximate* revivals whenever the two frequencies $\kappa_\pm + \omega$ are close to being commensurate (in which case the third frequency $\kappa_+ - \kappa_- = (\kappa_+ + \omega) - (\kappa_- + \omega)$ is also close to being commensurate), as apparent in Fig. 4(b). The revivals become exact in the special case of quenches to resonance ($\omega_0 = 0$), in which case the three frequencies reduce to two commensurate frequencies, $\sqrt{2\overline{g}^2}$ and $2\sqrt{2\overline{g}^2}$. In this scenario the environment spins oscillate periodically with a frequency that is half the oscillation frequency of the central spin, as illustrated in Fig. 5(b).

In all cases, the amplitude of these oscillations scales as $\mathcal{O}(1/L)$. This scaling can be understood by noting that $\sum_{j=1}^L \langle \psi_0 | S_j^z(t) | \psi_0 \rangle = 1 - \langle \psi_0 | S_0^z(t) | \psi_0 \rangle$, with the right-hand side being $\mathcal{O}(1)$. Because of the delocalization in the environment all contributions to the summation are on the same order, leading to the observed $\mathcal{O}(1/L)$ scaling.

To summarize, we note that semilocalization of the eigenstates can be observed by noting that an initial state that is supported on the localized component of the bright state (i.e. on the central spin) will lead to oscillations in $\langle S_0^z \rangle$ that do not vanish as the system size goes to infinity, whereas initial states that are supported on the delocalized environment spins exhibit oscillations that vanish in this limit.

# 7 Quenches to resonance with an unpolarized environment

In this section we consider generic quenches to resonance and use the known structure of the eigenstates to show that central spin observables do not relax to thermal equilibrium.

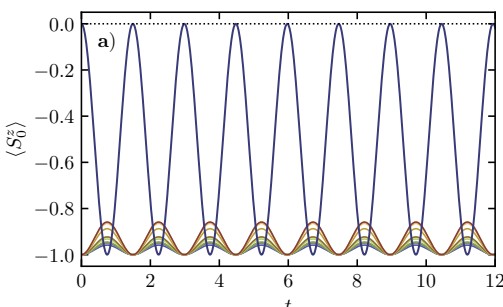
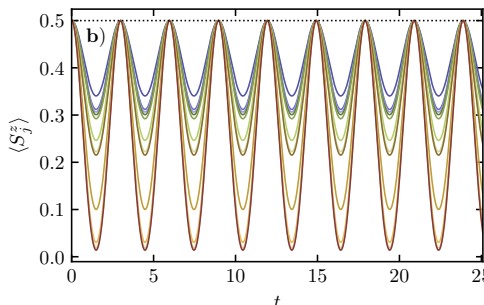

Figure 5: Quench dynamics in single-excitation sector for a quench to resonance ($\omega_0 = 0$). **(a)** Dynamics of the central spin magnetization $S_0^z$ for an initial product state that is either localized on the central spin (line starting at $\langle S_0^z \rangle = 0$) or on an environment spin, with the different lines starting at $\langle S_0^z \rangle = -1$ illustrating the $L$ different initial states. **(b)** Dynamics of the environment spin magnetization $S_j^z$ for an excitation initially localized on site $j$. *Parameters: $L = 12$, and $\tilde{g}_j$ is uniformly distributed in the interval $[1, 2]$ for environment spins with $s_j = 1/2$.*

Specifically, we consider

$$ S_0^z, \qquad P_0 = 1 - (S_0^z)^2, \qquad P_{\pm 1} = \frac{(S_0^z)^2 \pm S_0^z}{2}. \qquad (67) $$

with the latter two corresponding to projectors on central spin states $|0\rangle_0$ and $|\pm 1\rangle_0$ respectively.

While exact predictions in the inhomogeneous model are currently out of reach, we show that, for not too strong inhomogeneities, the late-time expectation values in the inhomogeneous model are well approximated by the diagonal ensemble expectation values in the homogeneous model. We refer to this approximation as the *homogeneous dephasing approximation* (HDA). Inhomogeneity in the couplings breaks the degeneracy of states in the homogeneous model, causing dephasing between formerly degenerate eigenstates. Meanwhile, the matrix elements of central spin observables between eigenstates are not significantly affected. A similar approximation was used in Ref. [71] for a nonintegrable Ising model in a *many-particle dephasing* regime. The HDA ignores any change to said matrix elements, and only accounts for dephasing.

Depending on the initial state and measured observable we can systematically probe the effect of bright states, both triple and double, as well as dark states. Specifically, we consider an initial state where the central spin is polarized in the state $|m_0\rangle_0$, the environment (which we again take to consist of spin-1/2 particles) is at infinite temperature in a fixed magnetization sector $M_E = M - m_0$, and the total magnetization[3] $M = 1$. The initial density matrix can be written as

$$ \rho(t = 0) = P_{m_0} \otimes \frac{\mathbb{1}_{M_E}}{\mathcal{Z}_E}, \qquad \text{with} \qquad \mathcal{Z}_E = \text{Tr}(\mathbb{1}_{M_E}) = \binom{L}{L/2 - M_E}. \qquad (68) $$

Here $\mathbb{1}_{M_E}$ acts as the identity on states with magnetization $M_E$ and as zero everywhere else. For convenience we take $L$, the number of environment spins, to be even.

## 7.1   Prediction of late-time values under the HDA

The HDA simplifies the prediction for late-time values of central spin observables by circumventing the use of the Bethe Ansatz solution, which is mathematically cumbersome.

---

[3]We are interested in probing the dark states, which are only supported in sectors with $M \neq 0$.

Within the diagonal ensemble, the late-time values of all projectors are determined by their overlaps with the eigenstates in the inhomogeneous model. Under the HDA, these overlaps are approximated by those with the corresponding states in the homogeneous model. This approximation can be justified by numerically comparing the expectation values of projectors in the homogeneous model with those of the inhomogeneous model. Fig. 6 shows that the eigenstate expectation values of $P_0$, $P_1$, $S_0^z$ and $J^2$ in the inhomogeneous model have small spread around the homogeneous limit.

**Initial state $m_0 = 0$.** We first consider the case where $m_0 = 0$ and hence $M_E = M$. The initial density matrix is only nonvanishing in the triple bright state manifold (35). The diagonal values are equal $\langle \mathcal{B}|\rho(t=0)|\mathcal{B}\rangle = 1/2$, as the triple states have half their weight on states with central spin value 0. The diagonal ensemble (obtained by setting the off-diagonal elements in the triple bright basis to zero) is thus

$$\rho_{\mathrm{DE}} = \frac{1}{\mathcal{Z}_E} \sum_{\mathcal{B}} \langle \mathcal{B}|\rho(t=0)|\mathcal{B}\rangle \, |\mathcal{B}\rangle\langle\mathcal{B}| = \frac{1}{2\mathcal{Z}_E} \sum_{\mathcal{B}} |\mathcal{B}\rangle\langle\mathcal{B}| \, . \tag{69}$$

The property $\langle\mathcal{B}|P_0|\mathcal{B}\rangle = 1/2$ also implies that the long-time expectation value of $P_0$ is

$$\mathrm{Tr}\left[\rho(t\to\infty)P_0\right] = \frac{1}{2\mathcal{Z}_E} \sum_{\mathcal{B}} \langle\mathcal{B}|P_0|\mathcal{B}\rangle = \frac{1}{2} \, , \tag{70}$$

where we further used that the total number of triple bright states is $2\mathcal{Z}_E$.

Under the HDA, we find for the projectors on central spin $\pm 1$ that (see Appendix B)

$$\mathrm{Tr}\left[\rho(t\to\infty)P_{\pm 1}\right] = \frac{1}{\mathcal{Z}_E} \sum_{J=|M|}^{L/2} C\left(\frac{L}{2}+J, \frac{L}{2}-J\right)\left(\frac{(c_{J,M}^{\mp})^2}{4(J^2+J-M^2)}\right) \tag{71}$$

$$\approx \frac{1}{4}\left(1 \pm \frac{8M}{L+2M}\right), \tag{72}$$

where

$$C\left(\frac{L}{2}+J, \frac{L}{2}-J\right) = \binom{L}{L/2-J} - \binom{L}{L/2-J-1}. \tag{73}$$

is the degeneracy of environment states with spin $J$, and Eq. (72) follows from Stirling's approximation in the limit $L \gg |M|$ (far away from the single-excitation limit).

**Initial state $m_0 = -1$.** For $M > 0$ (such that all dark state have central spin $|+1\rangle_0$), the initial state has overlap with both the double and triple bright states, but not with the dark states. In this case, the HDA gives:

$$\mathrm{Tr}\left[\rho(t\to\infty)P_0\right] = \frac{1}{\mathcal{Z}_E} \sum_{J=|M|}^{L/2} C\left(\frac{L}{2}+J, \frac{L}{2}-J\right)\left(\frac{(c_{J,M}^{+})^2}{4(J^2+J-M^2)}\right) \tag{74}$$

$$\approx \frac{e^{\frac{4M+2}{L}}}{4}\left(1 - \frac{8M}{L+2M}\right), \tag{75}$$

where the approximation on the second line again holds in the limit $L \gg |M|$. The additional exponential factor arises from the environment sector $M_E = M + 1$.

For the projectors on the central spin states $|\pm 1\rangle_0$ we similarly find

$$\mathrm{Tr}\left[\rho(t\to\infty)P_1\right] = \frac{2}{\mathcal{Z}_E} \sum_{J=|M|}^{L/2} C\left(\frac{L}{2}+J, \frac{L}{2}-J\right)\left(\frac{(c_{J,M}^{+})^2(c_{J,M}^{-})^2}{4^2(J^2+J-M^2)^2}\right)$$

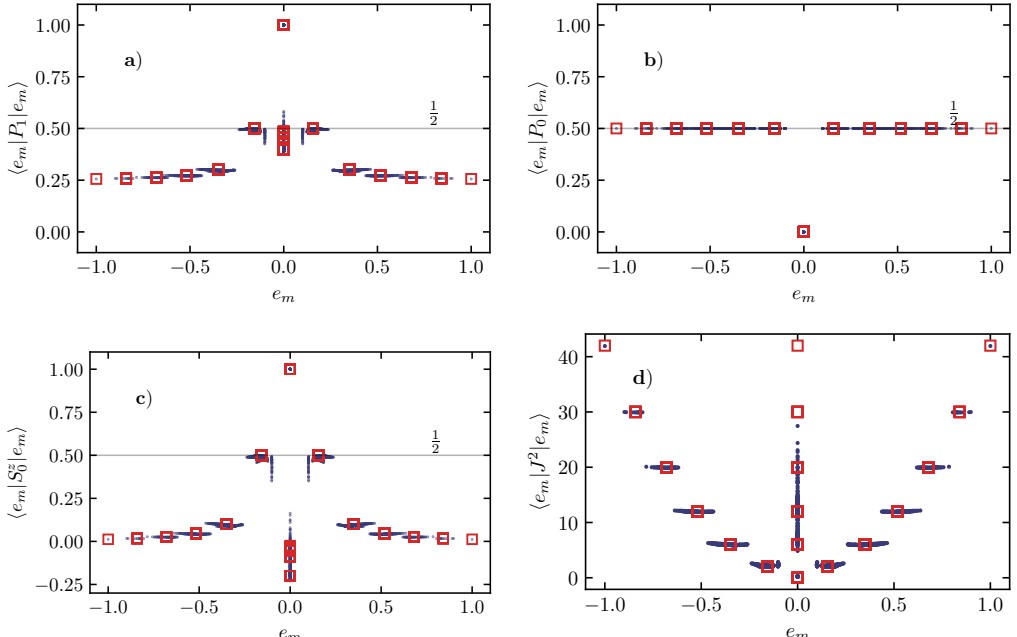

Figure 6: Eigenstate expectation values of: (**a**) $P_1$ and (**b**) $P_0$ (the projectors on central spin states as defined in Eq. (67)), (**c**) the central spin polarization $S_0^z$, and (**d**) the total environment spin $J^2$ (defined in Eq. (30)). The expectation values from the inhomogeneous model are plotted as dots, while those in the homogeneous limit are plotted as open squares. The energy eigenvalues $\{e_m\}$ are rescaled to lie between $\pm 1$. The plots show that upon introducing inhomogeneities in the XX couplings, the expectation values of central spin and environmental observables in eigenstates of the inhomogeneous model retain the overall structure and can be approximated by the corresponding values in the homogeneous model. *Parameters:* $L = 12$, $\omega_0 = 10^{-10}$, $g_j$ uniformly distributed in the interval $\pi/3 + [-0.5, 0.5]$, and $M = 1$.

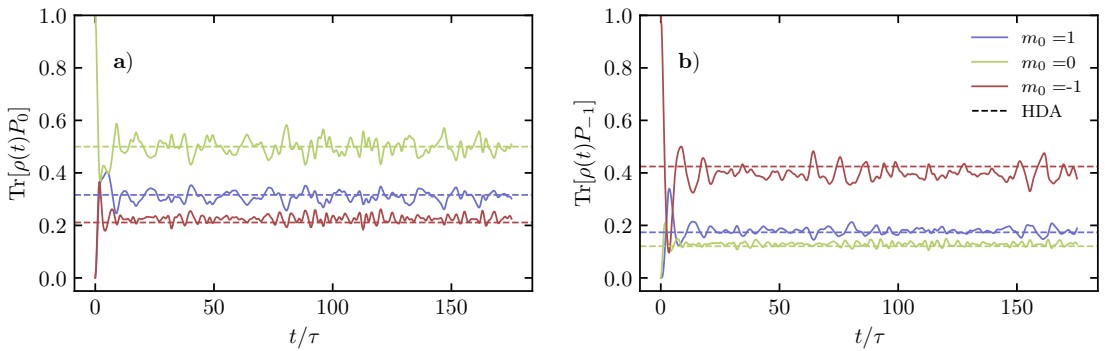

Figure 7: Quench dynamics for (**a**) $\mathrm{Tr}[\rho(t)P_0]$ and (**b**) $\mathrm{Tr}[\rho(t)P_{-1}]$. The dashed lines indicate the long-time values predicted by the HDA for different values of the initial central spin magnetization $m_0$. The projectors relax to their corresponding HDA values (dahsed lines), whereas in the Gibbs ensemble all solid lines would converge to a single value in the limit $L \gg |M|$. *Parameters:* $L = 10$, $\omega_0 = 10^{-10}$, $g_j \in \pi/3 + [-0.5, 0.5]$, $M = 1$, $\tau = (\sum_{j=1}^{L} g_j^2)^{-1/2}$.

$$+ \frac{1}{\mathcal{Z}_E} \sum_{J=|M|+1}^{L/2} C\left(\frac{L}{2} + J, \frac{L}{2} - J\right) \left(\frac{(c_{J,M}^+)^2 (c_{J,M}^-)^2}{2^2 (J^2 + J - M^2)^2}\right) \tag{76}$$

$$\approx \frac{3e^{\frac{4M+2}{L}}}{8} \left(1 - \frac{8M}{L + 2M}\right), \tag{77}$$

and

$$\mathrm{Tr}\left[\rho(t \to \infty)P_{-1}\right] \approx \frac{e^{\frac{4M+2}{L}}}{8} \left(1 - \frac{8M}{L + 2M}\right)$$
$$+ \frac{1}{4}\left[1 + \frac{8(M+1)}{L + 2(M+1)}\left(\frac{2M}{3M+2} + \frac{M^2}{(3M+2)^2}\right)\right]. \tag{78}$$

**Initial state $m_0 = +1$.** For $M > 0$, the dark states contribute to the quench dynamics.

Following the same steps as above, the late-time values are, for $L \gg |M|$,

$$\mathrm{Tr}\left[\rho(t \to \infty)P_0\right] \approx \frac{e^{\frac{4M-2}{L}}}{4} \left(1 + \frac{8M}{L + 2M}\right) \tag{79}$$

$$\mathrm{Tr}\left[\rho(t \to \infty)P_{-1}\right] \approx \frac{3e^{\frac{4M-2}{L}}}{8} \left(1 - \frac{8M}{L + 2M}\right) \tag{80}$$

$$\mathrm{Tr}\left[\rho(t \to \infty)P_1\right] \approx \frac{e^{\frac{4M-2}{L}}}{8} \left(1 + 3\frac{8M}{L + 2M}\right)$$
$$+ \frac{e^{-\frac{8M}{L}}}{4}\left[1 + \frac{8(M+1)}{L + 2(M+1)}\left(-\frac{2M}{3M+2} + \frac{M^2}{(3M+2)^2}\right)\right] + \frac{4M-2}{L+2M}. \tag{81}$$

The expressions above demonstrate that under the HDA, the late-time values of central spin observables retain memory of the initial state $m_0$. In contrast, the maximally mixed Gibbs ensemble in a fixed $M$ sector predicts the same late-time values for each central spin

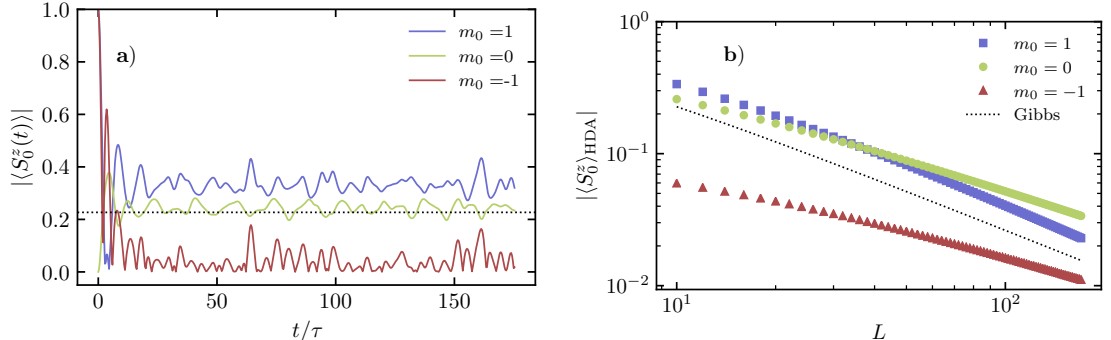

Figure 8: **(a)** Quench dynamics for $\langle S_0^z(t) \rangle$ for different initial values of $m_0$. The dotted line shows the value predicted by the Gibbs ensemble (82). **(b)** Expected late-time polarization under the HDA as a function of $L$ as compared to the Gibbs prediction (dotted line). Only the $m_0 = 1$ line has the same slope as the dotted line ($\approx 2.5/L$) at large $L$. *Parameters:* $L = 10$, $\omega_0 = 10^{-10}$, $g_j \in \pi/3 + [-0.5, 0.5]$, $M = 1$, $\tau = (\sum_{j=1}^{L} g_j^2)^{-1/2}$.

projector $P_{0,\pm 1}$, regardless of initial state[4] (up to $\mathcal{O}(|M|/L)$ corrections). For instance, in the limit $L \gg |M|$, we find that $\mathrm{Tr}\,[\rho(t \to \infty) P_0]$ approaches $1/4$ for $m_0 = \pm 1$ and $1/2$ for $m_0 = 0$, clearly differing from the Gibbs prediction of $1/3$.

## 7.2   Comparison with the inhomogeneous model

We can now compare the theoretical predictions in Sec. 7.1 with numerical results for the inhomogeneous model. In the case where $m_0 = 0$, the HDA prediction (70) applies exactly. This is because the initial density matrix is only non-vanishing in the triple state manifold (48), which has exactly the same weight on $|0\rangle_0$ and counting as the triple states in the homogeneous model.

Otherwise, the late-time values in the diagonal ensemble are different between the inhomogeneous and homogeneous models. Nonetheless, late-time values of central spin observables are well approximated by the HDA. These approximations are compared with numerical results for the inhomogeneous model in Fig. 7. In all cases the diagonal ensemble from the homogeneous model accurately reproduces the steady-state value of the inhomogeneous model. Furthermore, the late-time values for different initial states $m_0$ are clearly different.

Fig. 8(a) shows the corresponding dynamics of $\langle S_0^z(t) \rangle$. Crucially, in a given total magnetization sector $M$, the late-time expectation values heavily depend on the initial value of the central spin $m_0$ and differ from the Gibbs ensemble prediction, which can be calculated as

$$\frac{\sum_{M_E=M-1}^{M+1}(M - M_E)\mathcal{Z}_E}{\sum_{M_E=M-1}^{M+1}\mathcal{Z}_E} = \mathcal{O}(1/L). \tag{82}$$

This scaling with $L$ can be contrasted with the numerically observed scaling of $\langle S_0^z \rangle_{\mathrm{HDA}}$, as illustrated in Fig. 8 (b). While the $m_0 = +1$ curve shows the $1/L$ scaling from the Gibbs predictions, the $m_0 = 0$ and $m_0 = -1$ curves show different scaling exponents, approximately given by $L^{-0.8}$ and $L^{-0.7}$ respectively.

---

[4]The energy of the initial state is given by $\omega_0 m_0$. However, $\omega_0 = 0$ in a quench to resonance, hence the energy is always 0 and the use of the maximally mixed Gibbs ensemble is justified.

In the case where the initial environment does not have a fixed magnetization and is at infinite temperature, i.e. $\rho_E \propto \mathbb{1}$, one can also perform similar calculations for all $M$ sectors and perform a weighted average. The resulting long-time values for the polarization follow as $\langle S_0^z \rangle_{\text{HDA}} = \mathcal{O}\left(1/\sqrt{L}\right)$ for $L \to \infty$ as illustrated in Fig. 9(d). We note that this result is consistent with numerical results in the classical model (Fig. 10 in Appendix C.2), and inconsistent with the Gibbs prediction (82).

These results show that even in highly excited states, the integrability of the inhomogeneous model can be detected by the remnant memory of the initial state $m_0$ in late-time central spin observables.

# 8    Conclusion

We have established the integrability of the spin-1 central spin XX model by providing the exact construction of Bethe eigenstates and the extensive set of conserved charges, extending the results in Ref. [31] for the spin-1/2 central spin XX model. Like the spin-1/2 model, the eigenstates in the spin-1 model can be broadly classified into dark states and bright states, with the bright states showing *semilocalization* [67,70] in the single-excitation sector. Unlike the spin-1/2 model, bright states in the spin-1 model on resonance can further be classified into double or triple bright states.

The eigenstate structure of the spin-1 model prevents the central spin from reaching thermal equilibrium in quenches to resonance. In particular, for weakly inhomogeneous couplings, the late-time values of central spin observables approach diagonal ensemble expectation values in the homogeneous models. We expect this can be observed experimentally. Based on our numerical results, the time required to reach these late-time values is on the order of $10\tau$, where $\tau = (\sum_{j=1}^{L} g_j^2)^{-1/2}$. This is within the range of the spin relaxation time $T_1$ in NV systems, which limits the measurement of central spin observables in the $z$ basis. Indeed, the relaxation time we estimate as $10\tau$ is essentially the dephasing time $T_2$, which can be much shorter than $T_1$. For instance, at room temperature, $T_1$ has been observed to exceed 1 ms [72], while Ref. [73] recently measured $T_2 \approx 1$ $\mu$s.

In Appendix C, we have provided two pieces of evidence that support the integrability of the XX central spin model for any value of central spin $s_0$. First, the effective Hamiltonian at large $\omega_0$ for arbitrary $s_0$ is integrable. Second, numerical simulations of the fully classical model ($s_0, s_j \to \infty$) show residual memory in the late-time central spin magnetization, supporting integrability of the classical equations of motion. Within the truncated Wigner approximation, said classical equations govern dynamics for any value of $s_0$ [46,47]. An obvious future direction is to rigorously establish integrability at any $s_0$.

Other technical questions remain open. While the conserved charges in the spin-1 central spin XX model are closely related those of the XXZ Richardson-Gaudin integrable models, it is unclear how to incorporate them into the general framework of Richardson-Gaudin integrability. A specific challenge is the different scaling of the diagonal and off-diagonal terms in Eq. (23) with the number of environmental spins $L$. Because of this different scaling, it is unclear in which way the conserved charges constrain, for example, late-time observables in dynamical experiments.

# Acknowledgements

The authors thank H. Katsura, T. Skrypnyk, and A. O. Sushkov for helpful discussions and comments. This work was supported by: NSF Grant No. DMR-1752759, AFOSR

Grant No. FA9550-20-1-0235 (L.H.T., D.M.L. and A.C.); NSF Grant No. DMR-2103658, and AFOSR Grant No. FA9550-21-1-0342 (A.P.). A.C. and D.M.L. acknowledge the hospitality of MPI-PKS during July and August 2022 as part of the institute's visitors program. Numerical work was performed on the BU Shared Computing Cluster, using QUSPIN [74, 75].

## A Counting of states in the resonant homogeneous model

In this section, we provide the counting of the three classes of states (dark, double and triple bright states) in the homogeneous model on resonance.

The degeneracy of every eigenvalue is set by the total number of ways in which the $L$ environment spins can be combined to form a total spin $J$. We now focus on the case where each environmental spin is spin-1/2. For a given $M$, $J$ can then take integer values ranging from $J = J_{\min} = \max(0, |M| - 1)$ to a maximal value of $L/2$. Each of the spin-$J$ irreducible representations has multiplicities given by entries in Catalan's triangle

$$C\left(\frac{L}{2} + J, \frac{L}{2} - J\right) = \binom{L}{L/2 - J} - \binom{L}{L/2 - J - 1}. \tag{83}$$

Dark states reside in the $J_{\min}$ sector. Thus, for a fixed total magnetization $M \neq 0$, the degeneracy of the dark states immediately follows as

$$N_{\text{dark}} = C\left(L/2 + |M| - 1, L/2 - |M| + 1\right) \tag{84}$$

For bright states, the total number of double states is given by

$$N_{\text{double}} = \sum_{J=|M|+1}^{L/2} C\left(\frac{L}{2} + J, \frac{L}{2} - J\right) = \binom{L}{L/2 - |M| - 1} \tag{85}$$

in $M \neq 0$ sectors. The triple states are allowed in $J \geq |M|$ for $|M| \neq 0$, leading to

$$N_{\text{triple}} = 2 \sum_{J=|M|}^{L/2} C\left(\frac{L}{2} + J, \frac{L}{2} - J\right) = 2\binom{L}{L/2 - |M|} = 2\binom{L}{L/2 + |M|}. \tag{86}$$

It is easily checked that the total number of dark and (double and triple) bright states leads to the expected number of eigenstates in each magnetization sector.

For a given $M \neq 0$ sector, the ratio of the number of double states to that of the triple states is given by

$$\frac{N_{\text{double}}}{N_{\text{triple}}} = \frac{1}{2} \frac{L/2 - |M|}{L/2 + |M| + 1} \approx \frac{1}{2}\left(1 - 4\frac{|M|}{L}\right), \tag{87}$$

remaining finite in the limit of large $L$. The ratio of the number of dark states to that of the triple states is given by

$$\frac{N_{\text{dark}}}{N_{\text{triple}}} = \frac{2|M| - 1}{L - 2|M| + 2}, \tag{88}$$

In all cases, each class of states spans a nonvanishing fraction of the Hilbert space in the thermodynamic limit $L \to \infty$ provided $|M|$ scales with $L$. Keeping $|M|$ fixed and increasing $L$, the fraction of dark states vanishes as $\mathcal{O}(1/L)$.

For $M = 0$, the number of $J = 0$ dark states is given by $C(L/2, L/2)$. The number of double states is given by

$$N_{\text{double}} = \sum_{J=1}^{L/2} C\left(\frac{L}{2} + J, \frac{L}{2} - J\right) = \binom{L}{L/2 - 1} \tag{89}$$

in the $M = 0$ sector. There are triple states in the $J \geq 1$ sectors, resulting in

$$N_{\text{triple}} = 2 \sum_{J=1}^{L/2} C\left(\frac{L}{2} + J, \frac{L}{2} - J\right) = 2\binom{L}{L/2 - 1} = 2\binom{L}{L/2 + 1}. \tag{90}$$

The ratio $N_{\text{double}}/N_{\text{triple}}$ in this sector is exactly $1/2$, while

$$\frac{N_{\text{dark}}}{N_{\text{triple}}} = \frac{1}{L}. \tag{91}$$

## B  Expectation values at large system size under the HDA

In this section we detail the approximations used in evaluating the summations for the diagonal ensemble expectation values in the homogeneous model. In the limit where $J \ll L$, Stirling's approximation gives

$$C\left(\frac{L}{2} + J, \frac{L}{2} - J\right) \approx \frac{2J + 1}{L/2 + J + 1} \frac{1}{\sqrt{2\pi L}} \frac{1}{\sqrt{1/4 - (J/L)^2}} e^{Lf(1/2 - J/L)}, \tag{92}$$

where

$$f(p) = -p \log p - (1 - p) \log(1 - p). \tag{93}$$

Similarly, when $M_E \ll L$, the same approximation can be used for the ratio

$$\begin{aligned}
\frac{1}{\mathcal{Z}_E} C\left(\frac{L}{2} + J, \frac{L}{2} - J\right) &\approx \frac{2J + 1}{L/2 + J + 1} \sqrt{\frac{L^2 - 4M_E^2}{L^2 - 4J^2}} \exp\left\{\frac{2(M_E^2 - J^2)}{L}\right\} \\
&\approx \frac{2J}{L/2 + J} \exp\left\{\frac{2(M_E^2 - J^2)}{L}\right\},
\end{aligned} \tag{94}$$

to leading order in $J$. Evaluating diagonal ensemble expectation values involves summations of the form

$$\sum_{J=M_0}^{L/2} \left[\frac{M}{J^2 + J - M^2}\right]^p \frac{1}{\mathcal{Z}_E} C\left(\frac{L}{2} + J, \frac{L}{2} - J\right), \tag{95}$$

where $p$ is a non-negative integer. In all these cases, the summands are dominated by the smallest $J$, i.e. $M_0$. The summation may be approximated by substituting $J = M_0$ into the expression for the summand and multiply it by an $\mathcal{O}(1)$ factor to capture the entire sum. This factor can be extracted by comparing the sum to the summand for specific values of $M$, $M_0$, $M_E$ and (large) $L$. We find that multiplying the summand by a factor of 4 gives the overall best fit to the exact sum. This results in

$$\left[\frac{M}{M_0^2 + M_0 - M^2}\right]^p \frac{8M_0}{L/2 + M_0} \exp\left\{\frac{2(M_E^2 - M_0^2)}{L}\right\}. \tag{96}$$

For instance, substituting $M_0 = M_E = M$, $p = 1$, we recover one of the terms in (72). Figs. 9(a), (b) and (c) compare the HDA expectation values evaluated exactly and their respective approximations outlined in Sec. 7.1. They agree at large $L$.

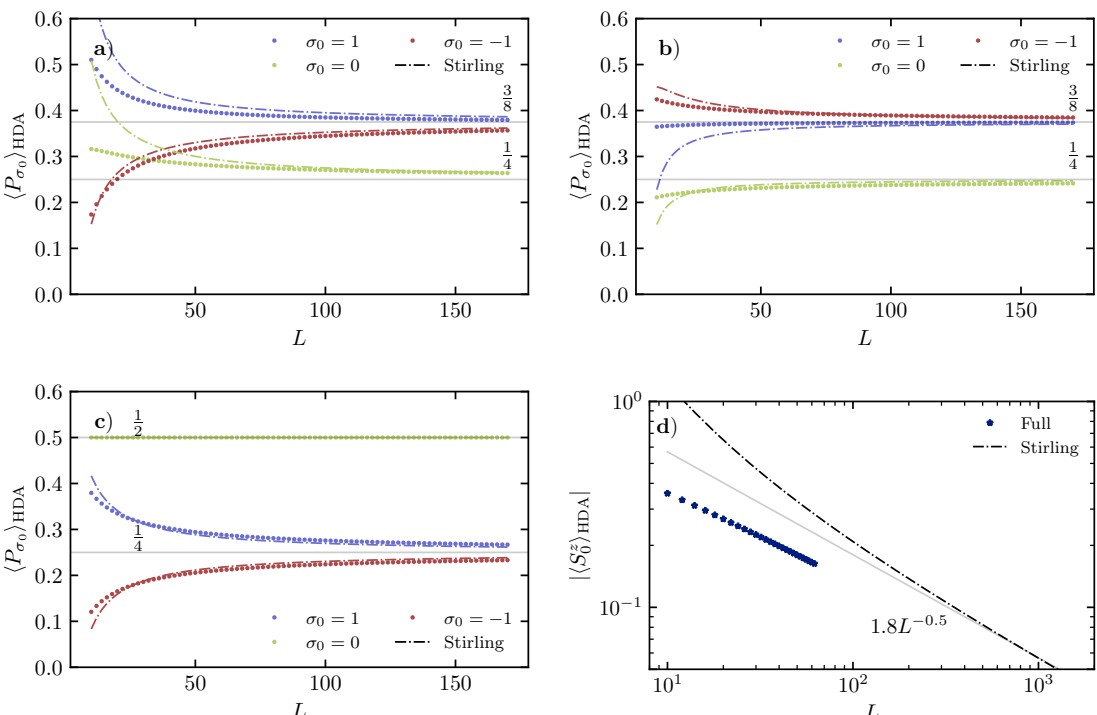

Figure 9: Quenches to resonance for a fully mixed environment. The HDA expectation values of the central spin projectors $P_{\sigma_0}$, where $\sigma_0 = 0, \pm1$ for initial states (**a**) $m_0 = 1$, (**b**) $m_0 = -1$, and (**c**) $m_0 = 0$ in the $M = 1$ sector as a function of $L$. The colored dash-dotted lines show the corresponding values under the Stirling approximation. (**d**) The expected remanent polarization averaged over all $M$ sectors when $|m_0| = 1$. Values computed with the exact expressions are plotted as stars, while those computed in the Stirling approximation are plotted as a dashed line. The solid line corresponds to $|\langle S_0^z \rangle_{\mathrm{HDA}}| \propto L^{-1/2}$. *Parameters:* $\omega_0 = 10^{-10}$.

# C    Integrability in higher spin models

Motivated by the results from the main text, we conjecture that the central spin Hamiltonian

$$H = \omega_0 S_0^z + \left( S_0^- G^+ + S_0^+ G^- \right), \tag{97}$$

is integrable for any central spin quantum number $s_0$. The integrability of this model has already been explicitly shown for $s_0 = 1/2$ in Ref. [31], and for $s_0 = 1$ in this work.

We support our conjecture through two pieces of evidence: perturbative conserved charges for $H$ in the limit of large $\omega_0$ (Appendix C.1), and numerical signatures of integrability in the classical limit of $s_0, s_j \to \infty$ (Appendix C.2). We also present a semi-classical argument for integrability, assuming exactness of the truncated Wigner approximation.

If the central spin Hamiltonian is integrable for all $s_0$, then it is likely that the related inhomogeneous Tavis-Cummings model (with off-diagonal disorder),

$$H_{\mathrm{TC}} = \omega_0 a^\dagger a + \left( a^\dagger G^- + a G^+ \right), \tag{98}$$

is also integrable. In Appendix C.3, we expand on this additional conjecture.

## C.1    Conserved charges far from resonance

A Schrieffer-Wolff transformation to $H$ in the $\omega_0 \to \infty$ limit provides an effective Hamiltonian (24)

$$H_{\mathrm{eff}}^{(1)} = \omega_0 S_0^z + \frac{1}{\omega_0} \left( S_0^+ S_0^- G^- G^+ - S_0^- S_0^+ G^+ G^- \right), \tag{99}$$

for any value of the central spin $s_0$. This Hamiltonian may be written as

$$\omega_0 S_0^z + H(\alpha) = \omega_0 S_0^z + \frac{1+\alpha}{2} G^+ G^- + \frac{1-\alpha}{2} G^- G^+ \tag{100}$$

where $H(\alpha)$ is a factorizable Hamiltonian from Eq. (14) and

$$\alpha = -\frac{1}{\omega_0}(S_0^- S_0^+ + S_0^+ S_0^-) \tag{101}$$

commutes with $S_0^z$, and so may be treated as a scalar in each $S_0^z$ sector. Thus, in each sector, $H_{\mathrm{eff}}^{(1)}$ is Richardson-Gaudin integrable and has an extensive set of conserved charges given by $Q_j(\alpha)$ (15). Combined with the fact that both the $s_0 = 1/2$ and $s_0 = 1$ cases are known to be integrable, the existence of this integrable limit is suggestive of integrability for any $s_0$.

Higher-order corrections to the effective Hamiltonian may also be computed, though it is unclear if they preserve integrability. We discuss them here for completeness and only note some connections with known integrable models.

The next correction to the effective Hamiltonian (at $\mathcal{O}(g^4)$) is given by

$$
\begin{aligned}
H_{\mathrm{eff}}^{(2)} &= -\frac{7}{12\omega_0^3} \Big( [[[S_0^+ G^-, S_0^- G^+], S_0^+ G^-], S_0^- G^+] + [[[S_0^+ G^-, S_0^- G^+], S_0^- G^+], S_0^+ G^-] \Big) \\
&= -\frac{7}{12\omega_0^3} \Big( 4(S_0^+ S_0^- S_0^+ S_0^- G^- G^+ G^- G^+ - S_0^- S_0^+ S_0^- S_0^+ G^+ G^- G^+ G^-) \\
&\qquad + 2(S_0^- S_0^- S_0^+ S_0^+ G^+ G^+ G^- G^- - S_0^+ S_0^+ S_0^- S_0^- G^- G^- G^+ G^+) \Big). \tag{102}
\end{aligned}
$$

Evaluating these corrections in a sector with fixed $s_0$ and $m_0$ results in linear combinations of four different operators,

$$G^-G^+G^-G^+, \qquad G^+G^-G^+G^-, \qquad G^+G^+G^-G^-, \qquad G^-G^-G^+G^+, \qquad (103)$$

where all terms of the form $G^-G^+G^+G^-$ and $G^+G^-G^-G^+$ cancel out. Each of these operators can again be shown to be Richardson-Gaudin integrable—although this does not guarantee that linear combinations will be integrable. $G^-G^+G^-G^+ = (G^-G^+)^2$ is the square of a factorizable Hamiltonian, as is $G^+G^-G^+G^-$. The terms $G^\pm G^\pm G^\mp G^\mp$ are also each integrable. To see this, we use the following generalization of a relation between Richardson-Gaudin charges introduced below Eq. (28),

$$(G^+)^k \left[(k+1)Q_j^{(-)} - (k-1)Q_j^{(+)}\right] = \left[(k+1)Q_j^{(+)} - (k-1)Q_j^{(-)}\right](G^+)^k, \qquad (104)$$

where $Q_j^{(\pm)} = Q_j \pm S_j^z$ and $k \geq 1$ is an integer. From this expression and its Hermitian conjugate, we see that a complete set of conserved charges for, say, $G^+G^+G^-G^-$ is given by

$$3Q_j^{(+)} - Q_j^{(-)} = 2Q_j + 4S_j^z, \qquad (105)$$

where $j \in \{1, \ldots, L\}$. Similar charges may be constructed for $G^-G^-G^+G^+$.

The two quartic terms that drop out in the effective Hamiltonian are the only terms that are not known to be integrable, such that the fact that they cancel out suggests that $H_{\text{eff}}^{(1)} + H_{\text{eff}}^{(2)}$ is itself integrable. Nonintegrability of $H_{\text{eff}}^{(1)} + H_{\text{eff}}^{(2)}$ does not imply that the higher-$s_0$ central spin XX models are nonintegrable—for instance, the same perturbative expansion holds for $s_0 = 1$, which is integrable. However, integrability of $H_{\text{eff}}^{(1)} + H_{\text{eff}}^{(2)}$ indicates a special structure in the full Hamiltonian, consistent with the conjecture.

## C.2   Classical limit

Taking $s_0, s_j \to \infty$ while keeping $g_j s_0 s_j \sim g_j^{\text{cl}}/2$ and $\omega_0 s_0 \sim \omega_0^{\text{cl}}$ finite results in a model which is formally identical to $H$,

$$H_{\text{cl}} = \omega_0^{\text{cl}} \tilde{S}_0^z + \sum_{j=1}^{L} \frac{g_j^{\text{cl}}}{2}(\tilde{S}_0^+ \tilde{S}_j^- + \tilde{S}_0^- \tilde{S}_j^+) \qquad (106)$$

$$= \omega_0^{\text{cl}} \tilde{S}_0^z + \sum_{j=1}^{L} g_j^{\text{cl}}(\tilde{S}_0^x \tilde{S}_j^x + \tilde{S}_0^y \tilde{S}_j^y), \qquad (107)$$

with the spins $\tilde{S}_0^\mu$ and $\tilde{S}_j^\mu$ being *classical* degrees of freedom.

If $H$ is integrable for all values of $\omega_0$, $g_j$, $s_0$ and $s_j$, it is natural to suspect that this limit model is also integrable. Conversely, integrability of the classical model suggests special structure with $s_j < \infty$. In this section, we numerically search for nonergodicity (a consequence of integrability) in the classical model (107), which can be simulated efficiently.

Equations of motion for the classical model are defined through Poisson brackets in the usual way:

$$d_t \tilde{S}_j^\mu = \{\tilde{S}_j^\mu, H_c\}, \quad \text{where} \quad \{\tilde{S}_j^\mu, \tilde{S}_k^\nu\} = \delta_{jk}\epsilon_{\mu\nu\rho}\tilde{S}_j^\rho, \qquad (108)$$

where $\epsilon_{\mu\nu\rho}$ is the Levi-Civita tensor and summation over the index $\rho$ is implied.

If the central spin is initially aligned along the $z$-axis while the environment is in an infinite temperature state, and a quench to resonance $\omega_0^{\text{cl}} = 0$ is performed, then ergodicity implies a late time average value of the central spin polarization given by

$$\lim_{T\to\infty} \frac{1}{T} \int_0^T dt \, \langle \tilde{S}_0^z(t) \rangle = \frac{1}{L+1}. \qquad (109)$$

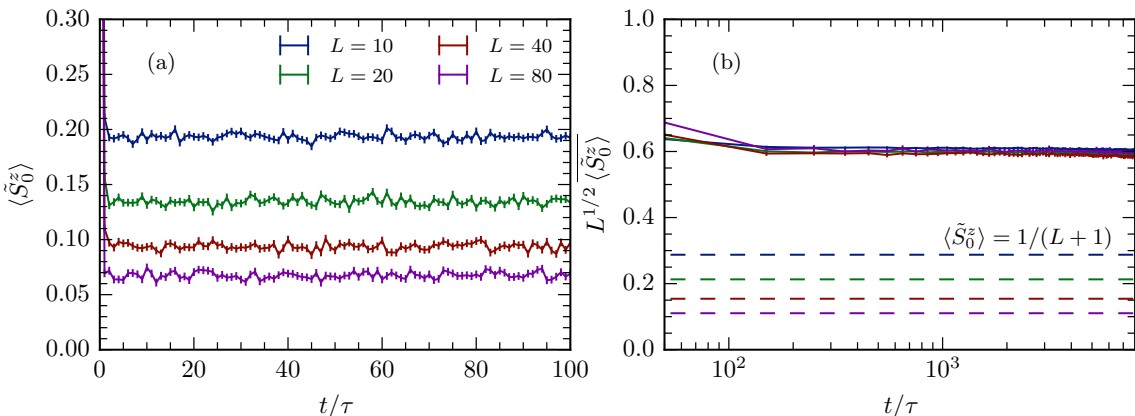

Figure 10: (**a**) The expectation value $\langle \tilde{S}_0^z(t) \rangle$ (110) in a quench to resonance of the classical central spin model (107) quickly reaches a steady state limit. (**b**) Rescaling $\langle \tilde{S}_0^z(t) \rangle$ by $\sqrt{L}$ collapses the late-time data, whereas the ergodic prediction scales as $1/L$ (dashed lines). *Parameters:* $\langle \tilde{S}_0^z(t) \rangle$ is computed from 200 samples of initial conditions in the integral Eq. (110) with $\omega_0^{\rm cl} = 0$ and $\tau = (\sum_{j=1}^{L}(g_j^{\rm cl})^2)^{-1/2}$. $\langle \tilde{S}_0^z(t) \rangle$ is further averaged over 200 realizations of $g_j^{\rm cl}$ drawn independently from a box distribution, $g_j^{\rm cl} \in \pi/3 + [-0.3, 0.3]$. In (**b**), $\overline{\langle \tilde{S}_0^z \rangle}$ is additionally averaged within bins of $100\tau$ to reduce its oscillatory part. Error bars give one standard error of the mean.

Here,

$$\langle \tilde{S}_0^z(t) \rangle = \int \left( \prod_{j=0}^{L} \frac{\mathrm{d}^2 \tilde{S}_j(0)}{4\pi^2} \right) 4\pi^2 \delta(\tilde{S}_0^z(0) - 1) \, \tilde{S}_0^z(t) \tag{110}$$

is the average of $\tilde{S}_0^z(t)$ over an ensemble of initial states with a fixed central spin state, and an infinite temperature environment.

Fig. 10 demonstrates that the late time average value of $\langle \tilde{S}_0^z(t) \rangle$ is *not* $1/(L+1)$. Instead, the late time value decreases more slowly with $L$, as

$$\lim_{T \to \infty} \frac{1}{T} \int_0^T \mathrm{d}t \, \langle \tilde{S}_0^z(t) \rangle = \mathcal{O}\left(1/\sqrt{L}\right), \tag{111}$$

numerically demonstrating nonergodicity.

The $1/\sqrt{L}$ scaling of the remanent magnetization is also a feature of the spin-1 model (see Fig. 8). That the same phenomenology persists in the classical limit favors the hypothesis of integrability of the classical model.

Integrability of the quantum model at any values of the central spin, including the classical large $s_0, s_j$ limit, also follows from semiclassical considerations. Namely, for this system one can anticipate that the truncated Wigner approximation (TWA) [46, 47] accurately describes dynamics in the large $L$ limit for any values of the central and environment spins $s_0, s_j$. This feature is general for all large $L$-models with long range interactions, where classical dynamics governed by Eq. (108) emerges as a saddle point within the path integral formulation of the Heisenberg evolution on a Schwinger-Keldysh contour (see for example Refs. [76–78]). Within the TWA the values of the spins are encoded in the Wigner function representing the initial state. Because the spin-1/2 or spin-1 systems are integrable, Eqs. (108)—which are expected to describe dynamics of the central spin in the large $L$ limit—must be non-ergodic as well to avoid thermalization.

Because these equations are independent of the spin quantum numbers, we can anticipate that integrability holds for all values of $s_0, s_j$.

These qualitative considerations cannot be viewed as a proof of integrability, as in general the limits $L \to \infty$ and $t \to \infty$ do not commute, and the TWA is expected to be accurate in the limit $L \to \infty$ first. The opposite limit is much harder to analyze analytically within the TWA and needs further study. Nevertheless, putting these subtleties aside, a combination of analytical and numerical evidence we presented in this paper suggests that the model is integrable in the limit $s_0 \to \infty$ for any $L$ and is integrable in the limit $L \to \infty$ for small values of $s_0 = 1/2, 1$. Because both of these limits are described by the same semiclassical equations of motion it is natural to assume that the model is integrable for any $s_0$.

### C.3   Inhomogeneous Tavis-Cummings model

Several interesting models may be obtained as limits of $H$. Assuming the integrability of $H$ for any values of $s_0$, $s_j$, $\omega_0$, and $g_j$, it is natural to suspect that the limit models are also integrable. One such model was the classical central spin model of Appendix C.2. Here, we remark upon another notable large $s_0$ limit, which results in an inhomogeneous Tavis-Cummings model.

Taking an alternative large-$s_0$ limit of $s_0 \to \infty$ with $g_j\sqrt{s_0} \sim g_j^{\mathrm{TC}}$ increases the central spin Hilbert space while maintaining its level spacing. The limit model above the ground state may be expressed as an oscillator model,

$$H_{\mathrm{TC}} = \omega_0 a^\dagger a + \sum_{j=1}^{L} g_j^{\mathrm{TC}}(a^\dagger S_j^- + a S_j^+). \tag{112}$$

The Hamiltonian $H_{\mathrm{TC}}$ is a generalization of the Tavis-Cummings model (which is known to be integrable [19, 79]) with inhomogeneous couplings. Due to its connection with the central spin XX model, we conjecture that the inhomogeneous Tavis-Cummings model is also integrable.

The Hamiltonian $H_{\mathrm{TC}}$ is known to be integrable when additional non-linear terms are introduced [80], but no $r$-matrix is known for the model without such additional couplings.

The consequence of integrability in the central spin-1 XX model is that the structure of the homogenous limit persists to large inhomogeneity of the couplings. We speculate that this is also the case in the Tavis-Cummings model—the inhomogeneous model continues to exhibit the phenomenology of the homogeneous model, such as a superradiant transition (as expected from mean-field calculations).

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
