# Peer review of "Integrability and quench dynamics in the spin-1 central spin XX model"

_SciPost Physics_

## Round 1 · Referee Report · Anonymous (Referee 1) · 2023-4-4

Strengths

Brand new construction of an integrable model.

Well written.

Detailed use of the integrability and the eigenstate structures to study the dynamics of the system and it lack of thermalisation.

A well argued and clear conjecture concerning a wider class of possible integrable models is provided.

Report

In this submission the authors, by building a set of commuting conserved charges and, via Bethe Anstatz, the explicit eigenstates of the spin-1 central spin model demonstrate the integrability of the model.

The classification of eigenstates into dark and classes of bright states combined with the counting arguments which insure the completeness of the Bethe Ansatz, provides them with a valuable way to tackle their dynamics. They study them in some level of details in the single excitation sector and they also provide numerical studies of various quenches for inhomogeneous coupling demonstrating the capacity of the HDA to predict long-time values of central spin observables.

The work is very well presented and is remarkably clear concerning the construction of both the conserved charges and the various types of eigenstates that one can encounter..

Arguments which favour the integrability of the model for arbitrary central spin (S > 1) are also presented and could extend to inhomogeneous Tavis-Cummings results. It is surprising that the construction does not immediately relate to the know large classe of integrable Richardosn-Gaudin models which results from well known diagonal r-matrices.

It is specifically through this last point that I feel the submission provides one of the acceptance criteria required for SciPost Physics. Specifically point 3:

“Open a new pathway in an existing or a new research direction, with clear potential for multipronged follow-up work”

By demonstrating the integrability of the spin 1 model and by showing that, contrarily to the spin 1/2 model, it
Is is not (at least not trivially) related to the known set of Richardson-Gaudin models, the authors open a up a w

Requested changes

None

---

## Round 1 · Referee Report · Anonymous (Referee 2) · 2023-4-12

Strengths

unexpected results
well-written

Weaknesses

The evidence provided to support the conjecture of integrability for arbitrary central spin overlooks important issues.

Report

This submission contains two main contributions about the spin-1 central spin model with XX exchange interactions.

The first is to prove integrability of the model within the well-studied Richardson-Gaudin framework. The motivation for this comes from a previous work that showed integrability for spin-1/2 central spin. The authors are able to extend this approach in a very comprehensive manner, one which includes the derivation of Bethe ansatz equations for the exact solution. This extension is not trivial, indeed the technique is not even obvious. This result in itself will be of interest to many. Their analysis also show that the notions of bright and dark states found for the central spin-1/2 system carry over to the central spin-1 system.

The second contribution is to use the exact solution to take a deep dive into the predictions of the model regarding physical behaviors. Primarily the focus is on gaining an understanding of thermalization properties. This is an important aspect for any of the several proposals to use central spin systems in quantum metrology and sensing as indicated in the manuscript.

In addition, the authors have included more than ten pages of supporting documentation by way of appendices.

My view is that the submission provides a novel and synergetic link between different research areas'' and thus meets a key criteria for publication in SciPost Physics. I am happy to support publication after the implementation of some
modifications as discussed below.

Requested changes

I request that the referencing is tightened up in a few places. After presenting Eq. (14), the authors cite several references. Among these are [34,35,48,49], but they are not appropriate. In [34,35,48,49], the coefficients of the $S_jS_k$ terms that appear in the conserved operators have the form
\begin{align*}
\frac{g_j^2+g_k^2}{g_j^2-g_k^2}
\end{align*}
while in the subsequent Eq. (15) of the manuscript they have the form
\begin{align*}
\frac{2g_k^2}{g_j^2-g_k^2}
\end{align*}
It was only some years after the studies [34,35,48,49] that the above non-skew-symmetric'' form appearing in (15) was observed in integrable structures, and the referencing should accurately reflect this property. It is actually crucial for the authors' construction to work, and therefore important that this feature, the absence of skew-symmetry, is not misrepresented through spurious references. The references [34,35,48,49] should be deleted from the above mentioned place in the text on page 8 (though they are appropriate for citation elsewhere in the manuscript).

With respect to the authors' own comment on page 8,

"It is an open question how the integrability of this model and the construction of the conserved charges can be incorporated in the general framework of (Richardson-Gaudin) integrability such as generalized Gaudin algebras [49], the algebraic Bethe ansatz [59], or constructions based on solutions to the generalized'' classical Yang-Baxter equation [44,60].''

answering the open question will require clear understanding of the nature of the problem. The reference [59] for the algebraic Bethe ansatz is a a rather generic one that bears little connection to the system under study. The manner in which the next sentence refers to the classical Yang-Baxter equation also gives the impression that the authors do not appreciate that the algebraic Bethe ansatz procedure arises as a _consequence_ of a solution of a Yang-Baxter type equation. It is more logical to first refer to the classical Yang-Baxter equation, then the algebraic Bethe ansatz. In relation to the algebraic Bethe ansatz and how it relates to this model, it will be more informative to cite the recent paper [S] discussing a modified Bethe ansatz approach.

In Appendix C, the authors offer two studies which they believe add weight to a claim that the central spin XX model is integrable for arbitrary central spin. The first of these deals with an effective Hamiltonian approach, while the second considers an analogous classical Hamiltonian. While I think that this is a useful study, I also believe that there are good reasons to discuss why integrability might fail for what are simple reasons.

With regards to the effective Hamiltonian, I have the following comment. It is entirely conceivable to have a situation where an integrable Hamiltonian is perturbed to yield a non-integrable Hamiltonian, yet both the Hamiltonian and the perturbation lead to the same effective Hamiltonian. To give an analogy, the Bose-Hubbard model is widely-thought to not be integrable, yet the field-theoretic Bose Gas limit (a.k.a non-linear Schroedinger model) is integrable. There are several ways to discretize the Bose Gas model to obtain an integrable system of bosons on a one-dimensional lattice, but none of these produce the Bose-Hubbard model, [KR,AK]. This example shows that drawing conclusions from an effective Hamiltonian is perilous.

Regarding classical integrability, the studies by Talalaev and collaborators [CRT,T] are particularly pertinent. They show that for the fundamental gl(n) Gaudin systems, naive quantization of the classically integrable systems does not work. To achieve quantum integrability requires certain quantum corrections'' that are highly technical to implement [T]. So, while I think that details of Appendix C can still be presented, I request that the authors make comment to also acknowledge that quantum corrections'' might be required for integrability to hold in arbitrary central spin XX models.

[S] T. Skrypnyk, Elliptic Gaudin-type model in an external magnetic field and modified algebraic Bethe ansatz Nucl. Phys. B {\bf 988}, (2023) 116102

[KR] A. Kundu and O. Ragnisco, A simple lattice version of the Nonlinear Schrodinger Equation and its deformation with exact quantum solution, J. Phys. A {\bf 27}, (1994) 6335.

[AK] L. Amico and V. Korepin, Universality of the one-dimensional Bose gas with delta interaction, Ann. of Phys. {\bf 314},
(2004) 496.

[CRT] A. Chervov et al, Rational Lax operators and their quantization, arXiv:hep-th/0404106.

[T] D. Talalaev, Quantization of the Gaudin system, arXiv:hep-th/0404153.

  • validity: high
  • significance: high
  • originality: top
  • clarity: good
  • formatting: excellent
  • grammar: excellent

Author:  David Long  on 2023-05-09  [id 3656]

(in reply to Report 3 on 2023-04-12)

We would like to thank the referee for their detailed report, helpful comments and positive assessment of our manuscript. In response to the comments:

  • We have tightened the citations as suggested. When introducing the conserved charges of the factorizable Richardson-Gaudin Hamiltonians we explicitly mention that different expressions exist for the conserved charges and provide separate citations to both the symmetric and the non-skew-symmetric constructions. Since the ‘symmetric’ conserved charges can be recast as the non-skew-symmetric ones through a redefinition of the interaction constant we have not removed the references to the former, but we have commented on this distinction and separated the references. When discussing the open question of how to relate this model to known constructions based on the Yang-Baxter equation we have added the provided reference [65] and added a reference to [63], which specifically considers the algebraic Bethe ansatz for Richardson-Gaudin models. While we appreciate that the algebraic Bethe ansatz also depends on a solution to the Yang-Baxter equation, we wanted to make the distinction between the algebraic Bethe ansatz, which starts from a solution to the Yang-Baxter equation and afterwards takes the quasiclassical limit, and other approaches that directly start from a solution to the classical Yang-Baxter equations. We have rephrased the sentence accordingly.
  • We fully agree with the comment on the effective Hamiltonian. Indeed, the integrability of effective Hamiltonians does not imply integrability of the actual Hamiltonian. We chose to investigate the effective Hamiltonian since the conserved charges of the spin-1/2 and spin-1 central spin models reduce to those of the effective Hamiltonian in the limit $\omega_0 \to \infty$ and since the block-matrix construction of the conserved charges similarly suggests a decomposition of the conserved charges in sectors with different central spin magnetization, as is done in effective Hamiltonians. We agree that this argument does not allow us to draw any conclusions on the integrability of the central spin-$s_0$ Hamiltonian, and do not wish to do so. We have added a short discussion to the manuscript pointing this out, while also noting that the effective Hamiltonian appears consistent with the structure of the conserved charges observed so far, and included the provided references.
  • We fully agree with the comment on classical integrability and have added the suggested comment. While it’s not possible to exclude the necessity of including additional terms in the classical Hamiltonian in order to obtain an integrable model, the fact that the classical dynamics appear to be nonergodic without such additional terms suggests that they may not be necessary.

---

## Round 1 · Referee Report · Jerome Dubail (Referee 3) · 2023-4-23

Strengths

  1. Carefully establishes the integrability of a non-trivial, relevant, quantum many-body Hamiltonian

  2. Conjectures the integrability of a whole new family of central-spin Hamiltonians with inhomogeneous couplings

  3. Very carefully written.

Weaknesses

None.

Report

The authors study a spin $s_0=1$ coupled to a set of arbitrary spins $s_j$ ($j=1,\dots, L$) with inhomogeneous couplings $g_j$, and establish the integrability of the Hamiltonian, by:

  1. mapping the different subsectors onto Richardson-Gaudin-type models known to be integrable (sections 3-4), and constructing the corresponding conserved charges

  2. explicitly constructing the eigenvectors

  3. studying the quench dynamics, and observing the absence of thermalization.

The paper is very well written and it includes a clear overview of the results in section 2.

The central-spin model with inhomogeneous couplings is a very natural model, which is potentially experimentally relevant. As far as I know it was not previously known that it is in fact integrable, therefore in my opinion this paper reports a very nice and potentially impactful discovery. I strongly recommend publication of the results in Scipost physics. The paper can be publised as it is. I only have a few suggestions/comments that the authors may want to consider, see below.

Requested changes

Here are a few suggestions that the authors may consider or not.

  1. The summary of the results in section 2 is focused solely on the technical results (exact formulas for charges and eigenstates etc.), but it says nothing about the more physical results of sections 6 and 7; perhaps some of these results could be highlighted in section 2 as well.

  2. The physical signature of 'semi-localization' as a non-vanishing oscillation of $\langle S_0^z \rangle$ at the very end of section 6 seems important to me, perhaps the authors should expand a bit this paragraph and highlight it more.

  3. About the conjectured integrability of the higher central spin $s_0$: wouldn't it be possible to support this conjecture numerically, by looking for instance at level statistics? If so, is there a reason why no level statistics is shown in the paper (it is a very standard signature of integrability, used in many contexts)? Also, the fact that the Tavis-Cummings model with inhomogeneous couplings could be integrable (as it is included in that family) sounds potentially important. Perhaps this fact deserves to be highlighted. Currently it is somewhat hidden in Appendix C.3.

  4. The caption of Fig. 8 is not quite clear, it gives some values of the parameters, in particular `$L=10$', but then Fig. 8.b displays a quantity where $L$ is varied. I suppose the parameters are only for Fig. 8.a, but this deserves to be clarified.

  5. In the conclusion, I find that the comment about the 'different scaling' of the matrix elements in Eq. (23) is a bit obscure. Looking at Eq. (23) it is not obvious that the matrix elements scale differently with $L$, and this does not seem to be discussed around Eq. (23). If the authors think this is important, then they should probably elaborate on this point.

Typos: - caption of Fig. 7, 'dahsed' -> 'dashed' - sentence at the end of section 6.2: 'we note [...] by noting' sounds weird - section 6.2, below Eq. (65): 'These dynamics' should probably be 'This dynamics'

  • validity: top
  • significance: high
  • originality: high
  • clarity: top
  • formatting: perfect
  • grammar: perfect

Author:  David Long  on 2023-05-09  [id 3658]

(in reply to Report 4 by Jerome Dubail on 2023-04-23)

We would like to thank Dr. Dubail for his detailed report, helpful comments and positive assessment of our manuscript. We appreciate the various suggestions, and have chosen to address all of them in the revised version of the manuscript.

1. We now also discuss our results on semilocalization and quench dynamics in Section 2.
2. We have highlighted this signature of semilocalization at the end of Section 6.2.
3. We thank Dr. Dubail for pointing this out. While we had performed such calculations, we neglected to include these in the manuscript. We have now included a calculation of the level spacing ratio statistics for a central spin-3/2 model in Appendix C, Figure 10, which exhibits the Poissonian statistics expected for integrable systems.
4. The caption has been changed accordingly.
5. We have chosen to remove the discussion of this scaling in order to avoid confusion, since it depends on the specific choice of parameters in the Hamiltonian, and a detailed analysis falls outside the scope of our paper.

These typos have also been corrected.

---

## Editorial Decision

resubmitted